# Li-Ion Battery Cathode Recycling: An Emerging Response to Growing Metal Demand and Accumulating Battery Waste

**Nikita Akhmetov [1], Anton Manakhov [1,*] and Abdulaziz S. Al-Qasim [2]**

[1] Aramco Innovations LLC, Aramco Research Center, 119274 Moscow, Russia
[2] Saudi Aramco, Dhahran 31311, Saudi Arabia
* Correspondence: anton.manakhov@aramcoinnovations.com

**Abstract:** Due to the accumulation of waste mobile devices, the increasing production of electric vehicles, and the development of stationary energy storage systems, the recycling of end-of-life Li-ion batteries (EOL LIBs) has recently become an intensively emerging research field. The increasing number of LIBs produced accelerates the resources' depletion and provokes pollution. To prevent this, the global communities are concerned with expanding and improving the LIBs recycling industry, whose biggest problems are either large gaseous emissions and energy consumption or toxic reagents and low recycling yields. These issues are most likely solvable by upgrading or changing the core recycling technology, introducing effective benign chemicals, and reducing cathode losses. In this review, we analyze and discuss various LIB recycling approaches, emphasizing cathode processing. After a brief introduction (LIB's design, environmental impact, commercialized processes), we discuss the technological aspects of LIB's pretreatment, sorting and dissolving of the cathode, separation of leached elements, and obtaining high-purity materials. Covering the whole LIB recycling line, we analyze the proven and emerging approaches and compare pyrometallurgy, hydrometallurgy, and cathode's direct restoration methods. We believe that the comprehensive insight into the LIB recycling technologies made here will accelerate their further development and implementation in the large-scale battery industry.

**Keywords:** Li-ion batteries; recycling; hydrometallurgy; metal extraction; cathode regeneration

## 1. Introduction

In 2019, the Nobel Prize for Chemistry was awarded to J. B. Goodenough, M. S. Whittingham, and A. Yoshino for the development of rechargeable Li-ion batteries (LIBs). Since the 1970s, the concept of a rechargeable battery has attracted many researchers including M. Whittingham [1,2]. He found the $TiS_2$ layered structure to reversibly insert lithium ions at high potentials demonstrating relatively high capacities. Then, other layer oxides in the charged state ($MoS_3$, $MoO_3$, $V_2O_5$, etc.) were explored and investigated as cathode materials in a pair with a lithium metal anode [3,4]. J. Goodenough was the first who reported already lithiated cathode material, $Li_xCoO_2$ (LCO), in the 1980s [5]. The use of the lithiated cathode allowed to substitute a pure Li anode with graphite which ultimately reduced the dendrite formation and finally led to the Li-ion rechargeable battery (LIB) commercialization by Sony in 1991. Since then, a large variety of layer-structured compounds of various compositions were developed—$LiNiO_2$ (LNO), $LiMn_2O_4$ (LMO), $LiFePO_4$ (LFP), $LiNi_xCo_{1-x}AlO_2$ (NCA), and $LiNi_xMn_yCo_zO_2$ (NMC) exhibit high working potentials of 4 V vs. $Li/Li^+$ and higher [6,7]. LIBs equipped with such cathode materials are still considered the most suitable battery solution due to the design flexibility, long lifetime, low self-discharge rate, and high energy density compared with other batteries [8]. That is why LIBs are used in almost every portable device and electric vehicle (EVs) aiming to decline $CO_2$ emissions and meet the principal international agreements (Kyoto Protocol, Paris Agreement, and UN Sustainable Development Goals) [9]. However, the life-cycle analysis of the well-to-wheel greenhouse gas (GHG) emissions for EVs and internal

combustion engines vehicles (ICEVs) indicated that the decrease of emissions varied from 10 to 50% depending on the many parameters, such as electricity source, battery type, etc. [10]. About 7% of the emissions from EVs are related to the production of batteries, a very energy-intensive process [11]. Moreover, eutrophication and terrestrial acidification, and other environmental impacts are also related to the mining of minerals and battery manufacturing processes. Hence, all means allow for the reducing of GHG emissions both related to LIBs production itself and EVs, in particular. Thus, recycling spent Li-ion batteries increases the environmental benefits of EVs [12,13].

It is not a surprise that over the last ten years, the demand for LIBs has drastically increased [14,15]. Nowadays, the total number of portable electronic devices (mobile phones, laptops, tablets) approaches 10 billion worldwide [16]. The LIB market exponentially increased from $12 to 50 billion in the 2010s and is expected to break through $77 billion in 2024 [17]. Another recent report forecasts that LIBs production will reach $116 billion in 2031 [18]. Such unstoppable and unprecedented growth will remain in the future due to the continuing energy transition and broad implementation of EVs [19]. Unfortunately, negative environmental consequences of such exponential growth are expected. If the large mass of LIBs is not correctly utilized (recycled), it will cause soil and groundwater pollution along with the depletion of natural resources. The biggest country extractors of lithium-contained ore are Australia, Chile, and China. The latter is the critical exporter of chemical-grade lithium compounds—carbonate and hydroxide [20]. Over 0.5 million tons of lithium materials were particularly produced in China in 2020 [3]. Overall, such a large-scale, instantly developing production calls for effective battery recovery strategies.

No more than 51% (data differ from source to source) of spent LIBs undergo recycling [21,22]—often polluting, energy-intensive, and inefficient recovery processes for environmentally harmful but valuable metals. The market size of LIBs positively correlates with the number of spent LIBs: 200,000 tons of ready-to-recycle LIBs in 2020 are expected to exceed 1,200,000 tons in 2030 [23]. Therefore, more productive, cost-efficient, and eco-friendly industrial recycling processes should be developed and commercialized.

In this review, we, from the chemical point of view, analyze the LIBs utilization technologies with a particular emphasis on cathode recycling. The current paper mainly reviews the most recent recycling trends (2020–2022). Throughout the review, we follow the logic to pay attention to every recycling stage in a universal manner and consider every technological option for each stage, whether it seems outdated (discussed briefly) or emerging (more details and analysis provided). Such a broad and topical coverage distinguishes this review from the others focused specifically on the recycling of EV cathodes [24], LiCoO$_2$ cathodes [25], all LIBs' components [26,27]; on the cells pretreatment [28] or on the commercialized technologies [29]. In other words, we suggest this review as a self-sustained, independent study that is enough for a primary and sufficient immersion into the LIB cathode recycling field. Indeed, this review covers the whole spectrum of the cathode recycling processes starting from cell discharging and finishing with battery-grade salt extraction. After an introduction (environmental impact, existing recycling processes, Section 1), we focus strictly on the LIB's processing methodology and the chemistry used within. We discuss the recycling structure, provide specific conditions and reagents, and analyze the advantages and limitations of each method (Section 2). Reasonable attention is devoted to the hydrometallurgical branch of recycling as one of the most promising routes for the end-of-life (EoL) LIBs treatment. Further, we proceed through the latest cathode restoration techniques and, finally, finish with a summary and perspective for the LIB cathode recycling. With the current review article, we would like to provide interested scientists, especially, novices to the LIB recycling culture, with valuable and hands-on theoretical and practical knowledge. We hope that the LIBs recycling technological aspects provided and discussed in this paper will enhance the search and then implementation of novel sustainable recycling routes.

### 1.1. Components of LIBs

To consider recycling processes deeper as well as existing industrial routes and countries' legislation, one should remember the principles of operation and components of rechargeable LIBs. LIBs are reversible systems and rely on the intercalation/deintercalation of lithium ions into the electrodes. Namely, during the charging process, $Li^+$ moves through an electrolyte from the cathode to the anode. Simultaneously, electrons go through an external circuit, and the cathode material (transition metal oxide) partially oxidizes transforming electrical energy into chemical. During the discharge, the intercalated to layered graphite lithium ions travel the reversed path, thereby supplying electricity to an energy consumer.

LIBs consist of a stainless-steel or nickel-plated shell, cathode, anode, and separator soaked in organic electrolytes (Figure 1). The cathode is typically represented as an active material (LCO, LMO, LNO, LFP, NMC) mixed with carbon and polymeric binder and cast on an aluminum current collector. The existing active materials vary in potential and capacity, so the final application dictates the selection. The content of active material is highly important—the higher amount increases specific capacity [30], whereas its lower content in favor of binder improves the interfacial resistance [31]. Thus, the cathodes' composition varies in a wide range: 60–95% of active material and 5–25% of carbon and binder material [32]. Carbon black, as the most commonly used conductive material, provides good electrical contact between the active material and the current collector. It also decreases the polarization of the electrode due to its large surface area and conductivity [32]. The binder material (PVDF and derivatives [33], polypropylene [34]) complements the cathode structure by linking the active and carbon materials with current collectors. Besides the support role, the current collectors are responsible for the passage of electrons through the external circuit. The chosen material should demonstrate high chemical and mechanical stability, high electrical conductivity, and low thickness [35]. The typically-used current collectors are made of copper for the anode and aluminum for the cathode. Special approaches such as coating with carbon or vanadium oxide as well as the plasma treatment are known to improve surface chemistry and electric conductivity of the final cathode [36].

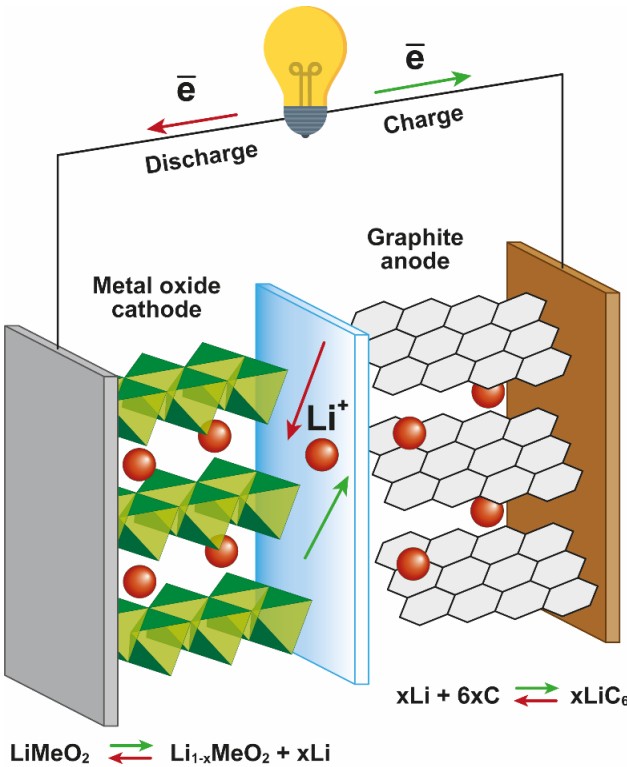

**Figure 1.** Schematic representation of a Li-ion battery cell.

The separator soaked in the liquid electrolyte is located between the electrodes, provides the charge balance, and prevents the battery from short-circuiting. Hence, the crucial requirements for a separator are excellent chemical stability, low ionic resistance and electronic conductivity, good wettability, low swelling, and uniform pore distribution [37]. Commercially available separators are based on microporous polymeric membranes, such as polypropylene (PP) [38], polyethylene (PE) [34], and fluorinated polymers (PVDF with co-polymers) [39]. At the same time, much research is now focused on finding more environmentally benign separators with no synthetic but recyclable polymers [40]. It is hard to avoid the novel tendency of developing gel- and solid-state electrolytes [41]. Such materials possess significant advantages compared to conventional liquid electrolytes in terms of safety—higher thermal stability and no leakage. However, solid-state electrolytes' application is still limited due to high bulk and interfacial resistances, electrochemical stability, etc. [42].

A correct choice of liquid electrolyte is crucial for a low-resistive interface responsible for the battery performance. The requirements for the electrolytes are environmental compatibility, low cost, electrochemical stability, and excellent battery performance. This is achieved by high ionic conductivity, low viscosity, and good wetting properties [43]. The electrolytes typically consist of an inorganic lithium salt (e.g., $LiPF_6$, $LiBF_4$, $LiClO_4$, $Li(SO_2CF_3)_2$) dissolved in non-aqueous solvents, such as propylene carbonate and mixtures of ethylene carbonate and aliphatic carbonate (dimethyl, diethyl, or ethyl methyl carbonate) [44]. These organic solvents are preferable due to their high electrochemical stability allowing the battery to operate in a high voltage range.

### 1.2. LIBs: Production, Cost, and Environmental Impact

For further understanding of the LIBs recycling specifics, one should consider the global distribution and current production rates of elements used in the cathodes—the most expensive part of the battery (Figure 2). Indeed, up to 75% of the total battery cost relates to materials and 50% corresponds to the cathode, specifically (Figure 2a,b).

Over 40% of lithium produced is implemented in LIBs [45]. Lithium as an element is stored in brines and pegmatites—aluminum silicate complexes, e.g., eucryptite, spodumene, and zinnwaldite; the other minor sources are mica, quartz, and feldspar [46]. Currently, Australia dominates the lithium minerals production market with 55 kilotons (kT) annually. The other leading country extractors are Chile (26 kT), China (14 kT), and Argentina (6.2 kT). The largest reserves are located in Chile—they exceed 9.2 Mt [45]. Despite the large explored—22 megatons (Mt)—and estimated—80 Mt—Li reserves, it is forecasted that the cumulative lithium demand will exceed the reserves by up to two times by 2050 [47], unless a regular and efficient recycling routine will be implemented. The cost of lithium (Figure 2c) is then driven by several factors: the ratio of demand/production, social and ecological issues in the country-extractors, and technology for lithium-contained minerals processing and LIBs production [48].

Another main element used in the LIBs cathodes, cobalt, is no less important. Except for LIBs cathodes (57% of usage), Co is consumed in alloy production, cemented carbides, and catalysts [49]. It was highly classified as the tenth most critical material in the world among 83 in 2021 [50]. As well as the lithium compounds, the production of cobalt is expected to increase by up to five times in 2025 compared with the start of 21st century (Figure 2d). Taking into account the exponentially developing EV market, Co deposits will be under certain pressure. Almost half (46%) of all explored Co reserves are concentrated in the Democratic Republic of Congo (3.5 Mt), and over 70% of its production is located there (120 kT) [45]. Other global cobalt suppliers are Russia (7.6 kT), Australia (5.6 kT), and Canada (4.3 kT). Such a storage and production centralization of Co bear serious risks to its continuous supply. Social and environmental issues such as political instability, warfare, and high pollution at the mining places are sad consequences of intensive mining in R.P. Congo [51]. UNICEF reported around 40,000 children were involved in cobalt mining and paid less than $2 per day [52].

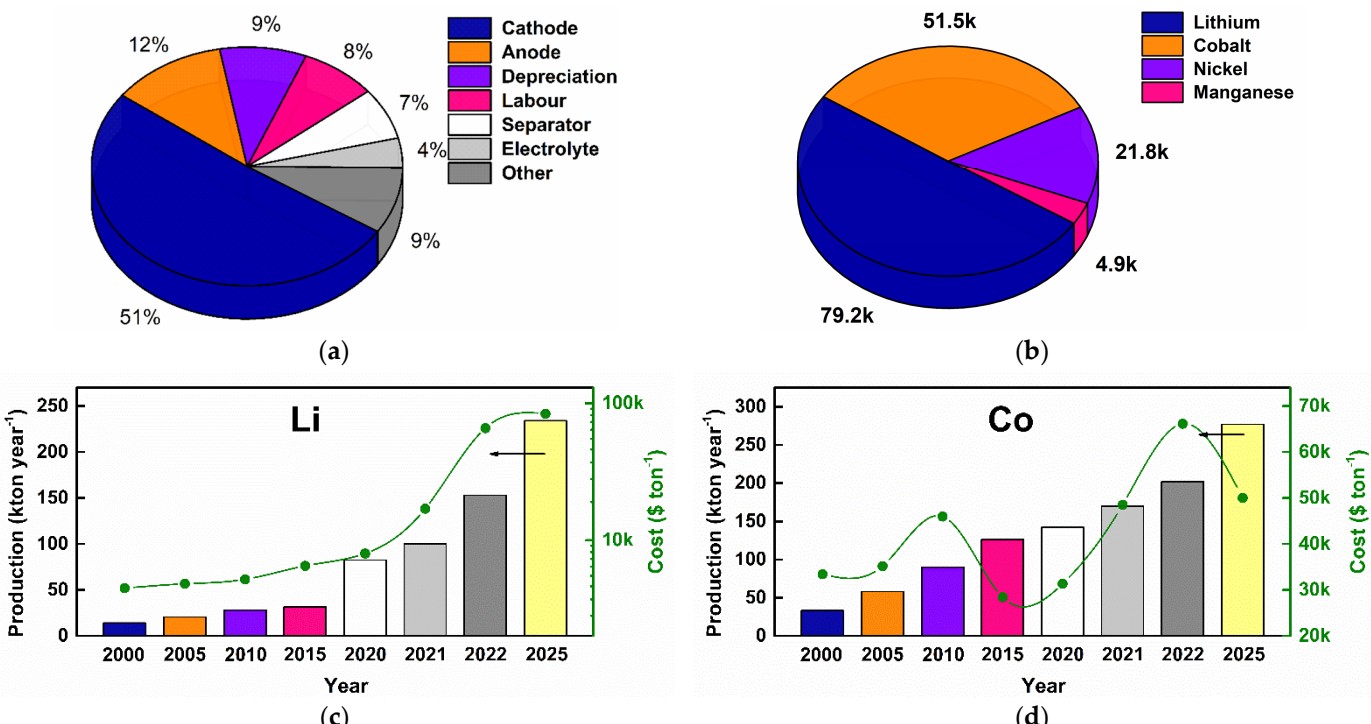

**Figure 2.** (**a**) Average cost specification for LIBs: components and service (data from [38,39]); (**b**) cost diagram of NMC cathode elements (in $ per ton): Li (>99.5% battery grade carbonate), Ni (>99.8%, ingots), Co (>99.8%, ingots), Mn (>99.7%, ingots) (data from [40–42]); (**c**) Li and (**d**) Co annual production (elemental, recalculated from ore mining) and cost in the 21st century: history and nearest forecast (data from [42–45]).

The situation with other battery materials such as nickel, manganese, aluminum, and copper is less critical compared to Li and Co. The main reason why the development of NMC, NCA, and other cathode materials attracts great attention nowadays is to reduce the reliance on cobalt. However, even Al and Cu recycling from spent LIBs has become a trend [53]. The explored reserves of Ni and Mn are approaching 95 and 1.5 Mt, respectively, with relatively insufficient annual production (2.7 and 0.2 Mt, respectively) [45], which stimulates the development of mixed-oxide cathodes.

The increasing demand for LIBs, geopolitics, social instability, etc. drastically enhances the price of battery-grade Li carbonate. Being at the level of $5k ton$^{-1}$ in 2010, the prize increased to $20k ton$^{-1}$ in 2021 and approached $82k kg$^{-1}$ in spring 2022 (Figure 2c). Cobalt had a more fluctuating tendency than lithium: the minimum price was around $28k ton$^{-1}$ in 2015 and raised to $66k ton$^{-1}$ in 2022 (Figure 2d).

LIBs are used in most electronic devices and recently started to compete with internal combustion engines (ICE) for domination in the automotive market. Worth mentioning is that ICE vehicle production releases fewer carbon oxides than EVs, but they compensate for it in the stage of usage. The valuable gain in terms of $CO_2$ emission will be even more if EVs are fueled with electricity produced by renewable green energy sources. It is estimated that EVs have already saved over 600,000 barrels of oil products per day and reduced greenhouse emissions by 53 Mt $CO_2$$^{-1}$ eq. in 2019 [54]. The EV sales equaled 2.1 M species in 2019 and 6.3 M in 2021, and are expected to approach 26.8 M in 2030 [54,55]. The sales are stimulated by government policies, for example, tax breaks and charging infrastructure. Of course, the invasion of EVs raises the problem of saving, and efficient and affordable recycling of spent LIBs [19].

It is interesting to observe that the total amount of LIBs used in EVs has been exceeding that for portable devices by more than two times (160 vs. 80 GWh) since 2020 and is forecasted to prevail by 15 times (2200 vs. 140 GWh) in 2030 (Figure 3a). It means that NMC-type cathodes used primarily in EVs are consequently displacing the LCO-based ones; sharing more than 80% of the LIB market in 2012, LCOs had a 9% fraction in 2021 and are expected to occupy less than 3% in 2030 (Figure 3b). It is not a surprise—LCO is not appropriate for EV applications due to the high cost of Co, safety issues (thermal runaway), and worse cyclability than that of NMC counterparts [56]. It also means that the LCO batteries are closer to the EoL state than NMCs, so a recovery strategy should be chosen accordingly. The number of waste LIBs from vehicles needed to be recycled can reach an optimistic value of 6.76 M in 2035 [57]. Overall, the growth of LIB production stimulates the proper and efficient recovery of cathode material.

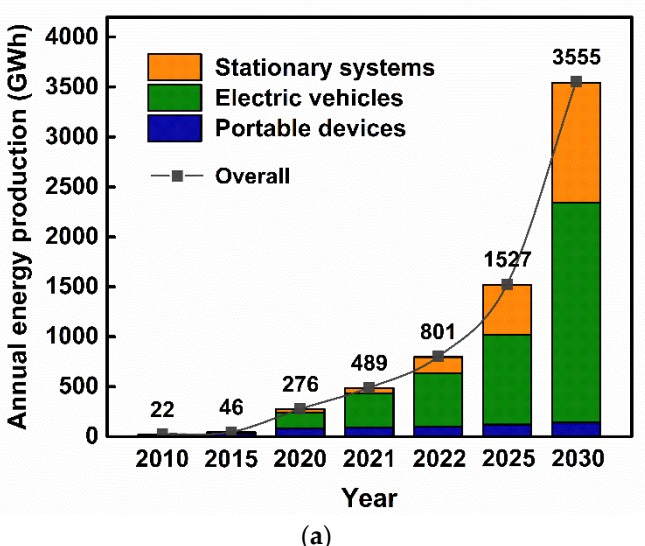
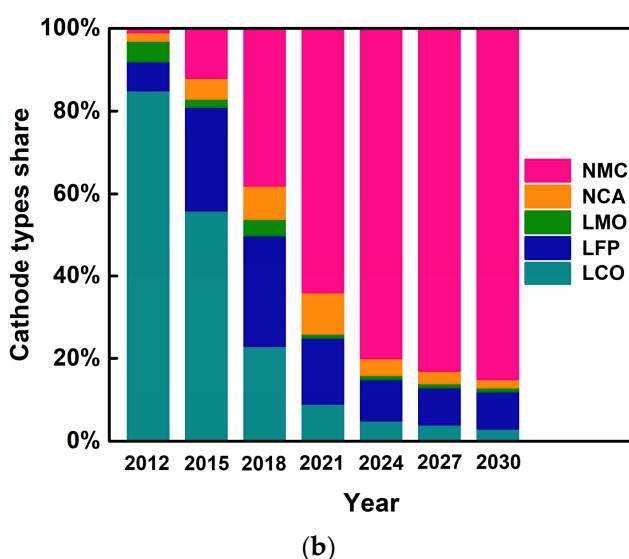

(**a**)                                                    (**b**)

**Figure 3.** History and forecast of (**a**) cumulative energy supplied by LIBs operated within portable devices, electric vehicles, and stationary energy storage systems (data from [58–61]) and (**b**) annual market share between cathode types (in installed GWh) (data from [62]).

### 1.3. Established LIBs Recycling Processes

Recent forecasts say that LIBs production will overcome 1 million tons annually by 2025 [20]. Starting with Toxco (1994) [58], the first commercial recycling line, today there are more than 32 established and planned recycling plants [29]. They already annually provide over 200 kT of LIBs recycling capacity (Figure 4) and are planning to install 70 kT additionally in the nearest future. Now, the largest facility (more than half of all) is provided by Asia (111 kT), followed by Europe (88 kT), North America (21 kT), and Australia (3 kT). The planned facilities will enhance the world recycling capacity to 400 kT [29].

In general, the entire recycling process can be separated into the following stages: pre-processing, mechanical, and pyro-/hydrometallurgical treatments [59]. The preprocessing stage is related to any process that does not affect the cell's structure, such as sorting out based on the cathode's nature and further discharging. The mechanical stage represents the techniques that do not involve the material chemical transformation, e.g., manual disassembling, shredding, liberation, the concentration of the material, and so forth [60]. Then, the obtained material is chemically treated using pyrometallurgy, hydrometallurgy, or a combination of them. Pyrometallurgy implements high temperatures to smelt the target metals [61], whereas hydrometallurgy represents the leaching of metal elements from the black mass with further precipitation using specific selective reagents [62]. Below, we will consider the most critical industrial processes more carefully, emphasizing publicly-available conditions, reagents, and other specifics.

Retriev, or Toxco, first developed for primary LIBs, is the first introduced recycling technology for rechargeable LIBs with an annual capacity of 4500 T and location in Trail, Canada [63]. The process starts with immersing the cells into the brine solution and their shredding. Alternatively, the crushing environment may be changed to liquid $N_2$, if the fraction of primary LIBs is high. After sorting out the plastic and small Al particles, the cathode-contained mass can be separated from a Li-rich liquid. The latter is then treated with $CO_2$ or $Na_2CO_3$ to precipitate $Li_2CO_3$ [64,65].

The collaborative recycling process of Sumitomo Metal Mining Company and Sony Electronics possesses an annual capacity of 150 T [63,66]. This pyrometallurgical process involves two calcination steps. In the first step, a heat treatment burns an electrolyte and cathode's organic additives. In the second step, calcination leads to the formation of an alloy of Fe, Ni, and Co. The final product after alloy leaching is CoO, which is later reused in LIBs manufacturing [66]. Unfortunately, as well as the Retriev process, Sumitomo–Sony emits a significant amount of exhaust gas due to the pyrometallurgical procedure which is, of course, a limitation.

The Recupyl process can be used for recycling both primary and secondary LIBs [67]. The Recupyl plant is located in Grenoble, France, and allows the processing of 110 T of spent batteries annually. At first, LIBs are mildly sheared in the inert, safe atmosphere of $CO_2$ or Ar. Then, after additional grinding and using a vibrating screen and magnetic separator (Fe removing), one obtains the fractions of different density and chemistry—plastic-paper (low-density) and Cu-Al-cathode material (high-density) fractions. Subsequently, Cu particles are removed with the 500 µm sieves, so the further operations deal with the undersize, electrode-containing fraction. This fine fraction is hydrolyzed with alkaline water, dissolving lithium salts and leaving heavy metal oxides and graphite in the solution. Filtered, they are leached with $H_2SO_4$ at 80 °C, and undissolved carbon is separated from the remaining solution. Finally, Li is recovered in the form of $Li_3PO_4$ by $H_3PO_4$, while $Co(OH)_2$ is precipitated by the addition of NaClO to the solution or by electrolysis.

The Umicore process possesses a significant capacity of 7000 T a year [63] and is focused on recovering Ni-MH and secondary LIBs. Here, the discharging and pretreatment stages are unnecessary. This method utilizes the complex shaft furnace with three compartments with different temperatures (300, 700, and 1200 °C [68,69]). Accordingly, each furnace section is responsible for the burning of an exact battery component. Firstly, at a low-temperature section, liquid electrolyte evaporates; then, pyrolysis of plastics takes place; and, finally, smelting of the recovering metals occurs. The obtained alloy undergoes hydrometallurgical leaching to obtain $Ni(OH)_2$ and $CoCl_2$ with the subsequent re-synthesis of $LiCoO_2$ [64]. In the Umicore process, much attention is devoted to the treatment of the emitted exhaust gases, which are recirculated and cooled in a special low-temperature chamber to suppress gaseous organics' formation (e.g., furans and dioxins) [70].

The Batrec process, which since the 1980s was focused on alkaline and Zn–C batteries, currently is used for LIBs recycling with an annual capacity of 200 T a year [71]. The battery cells initially stored and shredded in a $CO_2$ atmosphere are pretreated with moist air. After being neutralized, LIBs are crushed and prepared for the next operations. Unfortunately, no more specifications for this process are available.

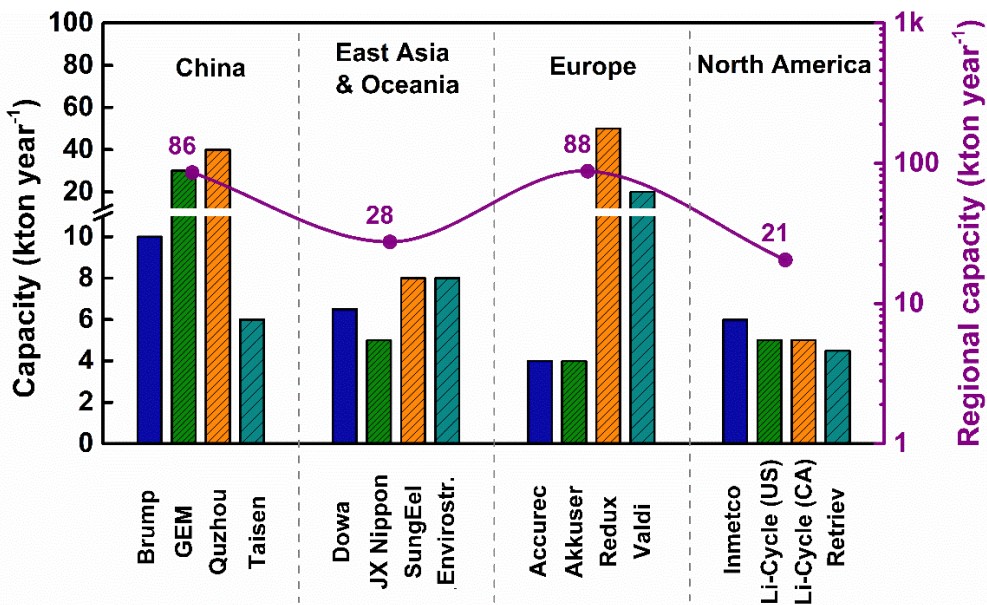

**Figure 4.** Annual capacity shares of currently available industrial LIB recycling processes (players with a production < 3000 kT year$^{-1}$ are not shown) and the summarized production values per region (data from [72]).

The Inmetco process (6000 T capacity, Elwood, IL, USA) utilizes special reducing pellets to be molten with the battery scrap and then refined in an electric arc furnace. The alloy of Ni, Co, and Fe are refined in the end [68,72].

Established in Switzerland (7000 T year$^{-1}$ capacity), the Glencore process recovers Ni, Co, and Cu using a combination of pyro- and hydrometallurgy. Unfortunately, no other metals can be extracted in the Glencore process [68,73].

The Accurec company (Krefeld, Germany), which recently added LIBs to their recycling capacity [69,74], can treat 4,000 tons of spent batteries annually. As in some previous technologies, the Accurec process utilizes a combination of mechanical and pyro- and hydrometallurgical treatments. After sorting and manual dismantling, the battery scrap is undergone at 250 °C and vacuumed to remove liquid electrolytes and light hydrocarbons [75]. Next, the processed feed is exposed to grinding, milling, and separation using a sequence of vibrating screens, a magnetic separator, and a zig-zag classifier. Then, the target fraction is sent to a series of pyrometallurgical treatments: a rotary kiln heated up to 800 °C is followed by an electric arc furnace, where graphite burning out accelerates the recovery of Co and Mn. The resulting Co-Ni-Mn-Fe alloy is further treated to mostly recover Co due to its much higher value, so other metals are lost in the slag phase [68]. However, due to the high concentration of Li in the slag, it can be recovered in the form of $Li_2CO_3$. After additional milling to a 100-μm size, the slag is hydrometallurgically treated with $H_2SO_4$, and then $Li_2CO_3$ is precipitated with a 90% recovery yield.

## 2. Technology of Recycling Li-ion Battery Cathodes

In the previous section, we have briefly discussed the environmental effect of LIBs, some established recycling processes, and countries' legislations. At the same time, much research is focused on upgrading the existing approaches and designing new methods of LIBs recycling. Here, we will speak about LIBs recycling more thoroughly, especially from the chemical point of view. We will consider and discuss all the recycling stages, starting from LIBs pretreatment and finishing with obtaining battery-grade salts ready for second-cycle battery production (Figure 5). Much attention will be devoted to the designed lab-scale routines, process conditions, and chemicals used by leading research groups specialized in LIBs recycling over the last decade. Looking ahead, it is important to note

there are recycling steps already suitable for use in the industry (e.g., pyrometallurgy; cell crushing; sieving, and separation of crushed fractions), other methods are currently being adopted to a large scale (inorganic acid leaching; metal ions separation; salt precipitation), whereas the rest part is emerging and capable only with the laboratory conditions (organic, alkaline, bioleaching; cell dismantling; cathodes direct regeneration; etc.). Throughout this section, we emphasize a correlation between an exact technology of a certain recycling stage and the acceptable application level, as well as discuss requirements and perspectives for the technology to be introduced into the industry.

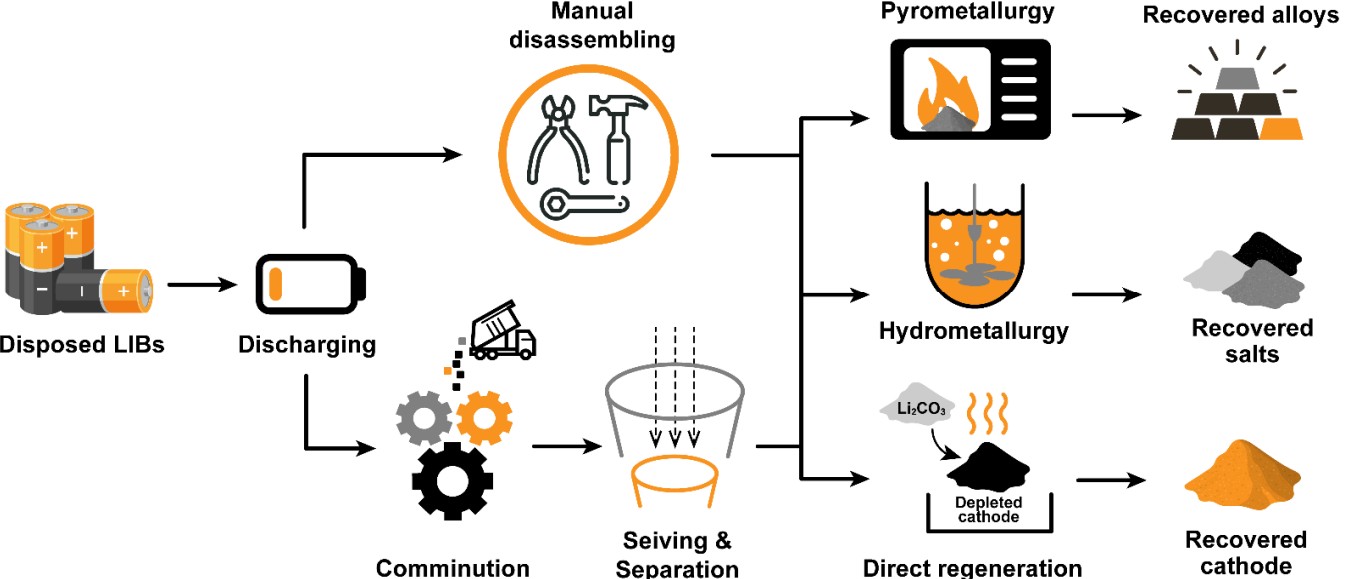

**Figure 5.** General scheme of LIB cathode recycling routes.

Special attention should be devoted to the low-temperature (LT) LIBs. Indeed, at temperatures below 0 °C LIBs generally face severe energy and capacity losses, impeded interfacial ionic transfer, and poor ions diffusion in bulk electrodes. In general, the main problems for LIBs at LTs are associated with the graphite anode (slow Li intercalation, high resistance of SEI) and liquid electrolyte (solidification, increased resistance). The anode-related issues are currently trying to be solved by the graphite oxidizing and metal nanoparticles' introduction (reduction of lithiation overpotential), particle size reduction, and structure and morphology modification (reduce Li plating). The spectrum of electrolyte improvement solutions is large and implies salt and solvent modification [76–78]. An efficient electrolyte for LT LIBs can be obtained by introducing F-contained chemistry (e.g., LiTFSI salt; fluoroethylene carbonate solvent additive, etc.) that would stabilize the SEI and provide the electrolyte with high ionic conductivity and low viscosity.

Improving cathodes is another path toward LT LIBs development. Currently, besides the enhanced impedance, at LT the cathodes suffer from dissolution and re-deposition of transition metals on the surface of the separator and anode [79]. The common solutions for the LT cathodes are surface coating (reduces contact resistance [80]), metal doping (improves the mobility of Li$^+$, stabilizes cathode-electrolyte interface (CEI) [81]), and particle size reduction (facilitates lithium insertion/extraction [82]). Overall, the modification of existing cathodes for LT LIBs requires a careful design of CEI by tuning the transition metal ratios and cathode surface pretreatment (e.g., by LiF-rich layers). In addition to the cathode modification of common materials (LCO, NMC, LFP), their substitution for emerging ones might be also beneficial for LT LIBs, such as $Li_3V_2(PO_4)_3$, $Nb_2O_5$, or Ni-based Prussian blue analogs [83].

Of course, both the cathode modification and the use of completely new materials call for severe modification of the recycling process of LIB as a whole and that of cathode materials, in particular. Such, the LiF introduction requires a responsible deactivation method during the recycling to avoid HF emission. The possible options are either selective capturing of the hazardous F-contained gases or LIB's components treatment in strong bases. e.g., NaOH [27]. The use of V-based cathodes additionally requires specific leaching conditions for the reductive/oxidative dissolution of vanadium. The sulfuric acid leaching solutions enhanced with a reductive agent ($H_2O_2$, $Na_2SO_3$) are capable to recover stable $V^{4+}$ in the form of $VOSO_4$ [84]. The use of expensive transition metals (e.g., Nb) as dopants or a main material bears additional economical challenges to the recycling of such cathodes. The losses of such high-cost materials should be minimized, otherwise, the recycling profitability will be questioned. Overall, the LT LIBs development and, particularly, their recycling technology, demands a deeper investigation in the nearest future and deserves a separate comprehensive overview. Meanwhile, in the current study, we are mostly focused on the recycling of traditional cathode materials: LCO and NMC.

LIB recycling can be separated into the "reuse" and "recovery" sub-processes. Such materials as Cu and Al (current collectors), Fe (case), and plastics can be directly reused after disassembly, and we will not devote much attention to them. We associate the "recovery" branch of recycling with processing a cathode material specifically, i.e., transforming the complex black mass to the high-purity chemical reagents. For convenience, we will join cell pretreatment techniques, namely, discharging, disassembling, and detaching active material, and will focus on all of them throughout the review.

### 2.1. Preparation of Spent LIB Cathodes for Recycling

2.1.1. Discharging

After the end of use, some power residue may remain in an LIB, which can cause short-circuiting and self-ignition due to exothermic reactions involving oxygen, water, and liquid electrolyte. That is why LIBs must be firstly discharged to at least 2.5 V before opening the cell [85]. Commonly, an EoL cell is immersed in the brine solution for 24 h or until complete discharge [63]. Although the method is relatively simple, the battery case tends to corrode under the saturated solution [86], and HF might be released. Alternatively, the recycling process can be performed in the Ar atmosphere or through cooling with liquid $N_2$—it significantly reduces the risk of explosion and toxic emission, so discharging might be unnecessary [24].

However, discharging with water solutions is still more affordable, so their investigation was the target for some research. LCO-contained LIBs were exposed to NaCl solutions of different concentrations (1, 5, and 10 wt.%) [87]. The higher the NaCl concentration was, the faster the cell discharged to 0.5 V; for 10 wt.% solutions, it took 20 min in contrast with 70 min for 1 wt.%. As stated above, the problem of cell corroding should be considered in terms of safety, and less-corrosive salts are to be implemented. Ojanen et al. studied the effect of stagnant and stirred NaCl, $NaSO_4$, $ZnSO_4$, and $FeSO_4$ solutions on the battery discharging behavior [88]. The NaCl solution demonstrated the fastest discharge, increasing with salt concentration. However, a slight chlorine emission was observed in the case of NaCl, whereas metallic precipitate was visible for the sulfates. Other researchers investigated alternatives to NaCl solutions: KCl, $NaNO_3$, $MgSO_4$, and $MnSO_4$ [89]. NaCl and KCl demonstrated fast discharge, but significant Fe cell case corrosion and electrolyte leakage. For $NaNO_3$ and $MgSO_4$, organic leakage was sufficiently smaller, although visually the corrosion took place. $MnSO_4$ showed the best safe properties (low corrosion and no leakage), but a slightly lower discharge rate compared with the other salts. Shaw-Stewart et al. largely expanded the number of known discharging salts [90]. Among some conventional examples (NaCl, $Na_2SO_4$, KCl, $K_2CO_3$, etc.), they investigated hydro-phosphates, sulfates, nitrates, nitrites, and hydroxides of $Na^+$, $K^+$, and $NH_4^+$. For the tested solutes, a large range of corrosive behaviors was detected from no corrosion to full destruction of the positive-side case. The $HCO_3^-$ and $H_2PO_4^-$ electrolytes showed the lowest corrosion

effect with the medium discharging rate. The NaNO$_2$ solution demonstrated unexpectedly interesting and prospective results—it provides an extremely high discharging rate with mild terminals' corrosion [90].

### 2.1.2. Manual Disassembly

After discharging, an LIB can be manually disassembled by traditional equipment, for instance, knives and saws. This relatively simple procedure requires essential safety concerns [57]. The emission of harmful volatile compounds detected after opening a cap did not exceed the critical values, as long as the cell is discharged, and ventilating pumps are on [91]. After breaking the battery, one can sort a plastic case, steel shell, Cu plate covered with graphite (anode), polymeric separator soaked in an electrolyte, and the target active material on Al foil (cathode). It is advised to rinse or wash with anhydrous ethanol the dismantled parts to minimize hazardous decomposition of the electrolyte and HF emission [27]. The plastics, metallic case, and separator are quite uniform and can be collected and directly sent to the specialized recycling plants. The manual disassembly allows for the selective dismantling of a valuable cathode, thus, simplifying further recovery and improving its efficiency [92].

The central problem of LIBs manual disassembling is the absence of automation. With the development of global mechanization, the automatic disassembly of EoL LIBs attracts much attention and effort. This approach would minimize the human workforce near a disassembly line as well as provide high selectivity of the extracted valuable material. Such an automatic disassembly methodology was proposed for opening LIBs pouch cells called Z-folded electrode-separator [93]. Hermann et al. reported that cell dismantling can be automated using a special jaw system, where the cell state can be continuously monitored [94]. At the same, there are still many issues limiting the implementation of the automatic disassembly line, such as different LIB designs and environmental impact.

### 2.1.3. Crushing, Comminution, and Separation

Alternatively, spent LIBs can be treated with the sequence of crushing, comminution, and separation processes. Currently, this method is more commonly applied in industrial-scale recycling due to its simplicity at the expense of efficiency. The manual disassembly and components separation does not address all the requirements yet to be scaled up [24].

Depending on a certain technology, spent LIBs should be either discharged, processed in an inert atmosphere, or treated under cooling with liquid nitrogen before crushing. For example, the Recupyl process is performed in an inert atmosphere and contains two steps: low-intensive two-blade rotary milling and high-intensive milling with a hummer [95]. Two approaches of wet and dry crushing of LCO-cathode battery were compared in [96]. In the wet case, water was fed into the crusher to form a slurry with fine particles that then got sieved. The dry crushing enabled the cathode and anode active materials to be detached from current collectors without damaging other LIB components that take place during the wet crushing. The mechanochemical reduction process involving a planetary ball mill was investigated to enhance the Li and Co extraction from the EoL LIBs [97]. After the variation of rotational speed, milling time, amount of the added grinding aid, etc., the authors obtained a positive shift in Co recovery efficiency. Wang et al. utilized cryogenic grinding as a comminution method to recover the cathode materials from the EoL LCO batteries [98]. In their set-up, the grinding tank was immersed in the tank with liquid nitrogen. The authors claimed that the low temperature (~77 K) allows them to selectively grind the cathode material, detaching it from the Al foil due to the PVDF glass transition and improved mechanical strength of Al.

Following the crushing and comminution steps, it is necessary to separate particles according to their nature. The sorting is implemented based on differences in the material's physical properties, namely, particle size, density, eddy current, electrostatic, and magnetic properties [92]. For example, in the Recupyl process, highly intensive magnets separate the steel components [99]. The finest particles are related to the cathode material, whereas the

shredded current collectors, casing, and plastics are left as a coarse fraction due to the more ductile nature of these materials [96,100].

The separation method based on the particle size of the crushed product is one of the most commonly used in LIBs recycling [101]. Traditionally, the milled LIBs scrap can be divided into three groups according to their size: <0.25 mm fraction rich in graphite and cathode active material, 0.25–2 mm fraction containing Al and Cu, and coarse fraction with Al and other components [100]. Wang et al. confirmed this distribution, investigating the content of LCO at different fractions obtained after sieving the battery scrap [102]. He found that <0.5 and 0.5–1 mm fractions contained 82 and 68 wt.% of Co, respectively. In the same study, the authors observed 92 wt.% Li-NMC material in the <0.5 mm fraction, which makes the separation by fractions quite effective for recycling pretreatment. Unfortunately, in the case of sieving, we are forced to choose between yield and purity of the cathode material [100].

An important issue of the separation stage is the hardness of graphite separation from the active material. He et al. successfully applied a froth flotation to sort LCO particles out of graphite [103]. The essence of the flotation separation method lies in the opposite surface wettability of graphite and LCO powder. In their experiments, the authors used the solution of $FeSO_4$ and $H_2O_2$ (Fenton's reagent) to remove a polymer binder and, thus, restore LCO's hydrophilicity. A similar but modified methodology was applied to the Li-NMC cathodes taken from commercial EVs. Extra hydrophobicity of the anode was provided by the addition of kerosene (collector) into the flotation process [104]. To enhance efficiency, the flotation parameters such as pH, collector type, frothier type, etc. can be varied [105]. He et al. have not stopped on that and expanded the method by combining it with heat and ultrasonic treatment. The limitation of the primary method relied upon the formation of $Fe(OH)_3$ film on the LCO surface which reduced flotation efficiency [106]. The introduced heat treatment improved the LCO grade to 90%. Then, to avoid side pyrolysis products, the following ultrasonication was added [107]. Under optimum conditions, 450 °C for 15 min, He et al. achieved 94 and 97% LCO grade and recovery rate, respectively.

Another efficient separation approach is the electrostatic method, which is based on different material's behavior on the electric field. The separation set-up consists of two types of electrodes—ionization and static. Briefly, the ionization electrodes when active pin non-conductive particles, while the static ones attract the conductive species. The particles that are too heavy to be pinned are collected as middling. After the condition's variation, it is possible to achieve an LCO purity higher than 95 wt.% [108].

Completely another approach was represented by initiating the eddy current in the mixed shredded battery powder. It allows the conductive materials (e.g., Cu, Al) to be separated from the non-conductors under scrap exposure to an alternating magnetic field [109]. Marinos et al. applied this concept to collect Co and Li as a non-electromagnetic fraction from the mass of LCO-contained batteries [110]. Zhang et al. expanded this method: with the help of magnetic and pneumatic separators, they succeeded to remove the metal shell and separator from the LCO LIBs scrap [111].

One more prospective method is ultrasonication used to detach the cathode active material from Al current collectors. Li et al. examined simple agitation, ultrasonication, and the combination of them to separate LCO from the Al substrate [112]. They found out that these methods are effective only when they are applied together. Such a combination does not require any hazardous chemicals during the detaching procedure (e.g., DMF), hence, can be considered as more environmentall friendly.

The separation of non-magnetic materials based on a difference in densities is implemented, for instance, in the Recupyl process [113]. Another example is the separation of Cu (8.96 g cm$^{-3}$) and polyethylene separators (0.9 g cm$^{-3}$). Using a high-density media, diiodomethane (3.3 g cm$^{-3}$), after 0.5 h exposure the current collector and separator can be separately collected from the bottom and surface of diiodomethane, respectively [114].

## 2.2. Types of Cathode Recycling Processes

Currently, the LIBs cathode recycling approaches are classified into two major methods—pyrometallurgy and hydrometallurgy. Pyrometallurgy implements several stages of thermal treatment (300–1000 °C) led to removing polymeric binder and conductive carbon additives and sintering the target cathode metals. Hydrometallurgy relies on chemical treatment in removing the binder and detaching the cathode material from the current collector. Hydrometallurgical treatment implies the dissolution of cathode material (LCO, NCA, NMC, etc.) by chemical reagents transferring the valuable metals to a mother solution in the form of ions. The metal ions are then separated and precipitated as commercial-grade, high-purity salts. Often, the combination of pyro- and hydrometallurgy is utilized on a large scale, because each method separately possesses its strengths and limitations. In this section, we will consider both methods and analyze recent research affords to create an efficient procedure for LIBs cathode dissolution.

### 2.2.1. Pyrometallurgy

A pyrometallurgy has first come to the scene for the recycling of primary batteries. Zn, Ni, Cd, and other metals can be recovered from spent Ni-Cd and Zn-Mn batteries [115]. During pyrometallurgical thermal treatment, plastics, binders, and electrolytes decompose, whereas metal oxides form alloys and move into a smelt slag. The pyrometallurgy method is widely used in industry due to its simplicity and adaptability—it does not require any special LIBs pretreatment. Pyrometallurgy implies a several-step thermal treatment to consequently remove volatile compounds. Such a two-step approach was applied by Fouad et al. [116]. The LCO electrode materials firstly were treated at 150–500 °C for 1 h to remove organic compounds, primarily, polymeric binders. Then, by increasing the temperature to 700–900 °C and holding the samples for 1 h, carbon, and residual organics were burned. Finally, the LCO mass was treated with $Fe_2O_3$ at 1000 °C for 4 h enabling the regeneration of Li/Co material in the form of submicron particle ferrite compounds.

To illustrate the thermal behavior of the cathodes, Yang et al. performed the thermogravimetric analysis with mass spectrometry (MS-TGA) [117]. They identified the first weight loss to occur at ~500 °C. It was related to the PVdF decomposition due to the intensive emission of $CO_2$ and $H_2O$ in the gas phase. The second MS-TGA peak (associated with $CO_2$) appeared at the range of 500–650 °C. The authors claimed this emission to be the interaction between the acetylene black binder and the cathode active material that led to the reduction of transition metal. Moreover, pyrometallurgy was successfully combined with the flotation separation method to utilize the spent LCO material [118]. The loss analysis studied with TGA showed several peaks. The low-temperature emission took place at 30–150 °C and corresponded to the evaporation of the residual liquid electrolyte. The 450–550 °C temperature range was related to the organic binder decomposition.

Diaz et al. compared traditional and microwave-assisted pyrolysis in treating NCA cathodes [119]. The 600 °C conventional pyrolysis completely removed the organic binder, but insufficiently detached the cathode material from Al foil. The applied microwave pyrolysis diminished the number of heavy gases by decomposing them into short-chained molecules. Interestingly, the microwave-assisted pyrolysis initiated a catalytic steam reforming reaction that led to the $H_2$ and CO formation. The optimum pyrolysis temperature considering possible toxic emissions and yields was established to be 350 °C. Another way to lower the temperature of pyrolysis and detach the active material from the substrate more effectively is the use of special additives such as CaO [118]. Their main purpose is to promote the decomposition of polymeric binders and enhance the interaction of other system components with each other. In these experiments, the NMC cathode was cut into small pieces and covered with CaO powder on both sides. The authors observed significant weight loss at 300 °C that was attributed to PVDF decomposition, due to the known high stability of NMC and CaO at such temperatures. At temperatures above 300 °C, the weight loss was ascribed to the reduction of CaO and acetylene black.

The approach to recover $Li_2CO_3$ from the EoL LIBs under vacuum was described by Xiao et al. [119]. The vacuum pyrolysis was carried out in oxygen-free conditions and allowed to extract $Li_2CO_3$ and manganese oxide decomposing the PVA binder into the gaseous compounds. Vacuum pyrolysis was also utilized by Sun et al. [120]. They put the LCO cathode material into the vacuum furnace without any crushing pretreatment. The authors stated 600 °C, $10^{-2}$ Bar, and 0.5 h of burning to be enough for the complete peel-off of the cathode material. At higher temperatures, Al lost its ductility which made the further sorting out much harder. The analogous pyrometallurgical treatment at atmospheric pressure showed oxidation of the cathode that made it breakable. Continuing the illustration of cathode materials regenerating, Li et al. investigated the reaction of LCO with graphite in the oxygen-free media at 1000 °C [91]. As a result, they achieved a rate of recovery of Li (in the form of carbonate), graphite, and Co of 96%, 99%, and 91%, respectively. Overall, the pyrometallurgical technologies are indeed quite simple and scalable, but energy-consumptive and demanding for the equipment.

### 2.2.2. Hydrometallurgy

A hydrometallurgy of LIBs cathodes is based on their chemical (instead of thermal) treatment. Hydrometallurgy is designed to suppress large energy consumption and toxic gaseous emissions distinctive to pyrometallurgy. Hydrometallurgy comprises several consequent stages of cathode handling: battery discharging and disassembling, cathode pretreatment (detaching the active material; binder removing), dissolution of active material (leaching), selective metals extraction, and/or salt precipitation (Figure 6). All these steps correspond to the whole process of cathode recycling—from LIBs collecting to battery-grade salt extraction ready for the cathode re-synthesis or other high-cost material fabrication. The first part of hydrometallurgy implies LIB disassembly and cathode preparation. It varies from lab-scale methods (manual disassembling/unwinding/cutting) to already-applied industrial processes, e.g., controllable crushing and sieving. We thoroughly discussed the pretreatment methods in Section 2.1, so we are switching to active material detaching, binder removal, and so forth.

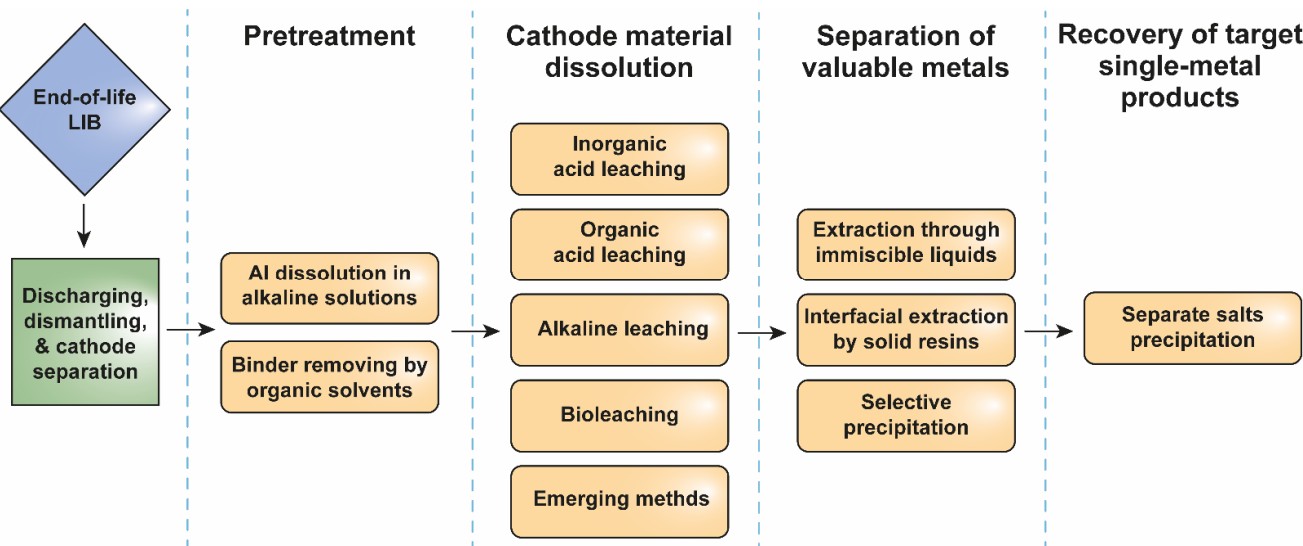

**Figure 6.** Common roadmap for the hydrometallurgical process of LIB cathodes recycling.

#### Active Material Detaching and Binder Removing

In addition to thermal treatment (pyrometallurgy), there are several approaches to chemically separate a cathode active material, polymeric binder, and Al current collector from each other. These are binder dissolution in the specialized solvents and dissolution of Al substrate in the strong bases. The most common solvent that can be used for the binder

dissolution is N-methylpyrrolidone (NMP) due to the high boiling point (200 °C) and PVdF solubility of up to 200 g per kg of solvent at 100 °C [121]. N,N-dimethylacetamide (DMAC) possesses acceptable PVdF solubility as well (100 g kg$^{-1}$ solvent) and was used to substitute less economically affordable and more toxic NMP [122]. After establishing the suitable solid-to-liquid ratio (1:5), the authors effectively separated the LCO and carbon additives of the solvent. Besides its low cost, DMAC can be easily recycled after usage by being evaporated at ~120 °C for 12 h [122].

A promising idea was proposed by Yang et al. to apply an ultrasonic treatment together with a solvent [123]. An introduction of the sonication allowed the avoidance of high temperatures (>100 °C) and promoted the dissolution of the binder due to the cavitation effect. The authors investigated a decent set of solvents, namely, NMP, CCl$_4$, CH$_2$Cl$_2$, acetone, and water. The high dissolution rate was observed only for NMP (99%), whereas it does not exceed 18% for the other solvents. Another set of chemicals, i.e., DMF, DMAC, N-N-dimethyl sulfoxide (DMSO), ethanol, and NMP, was evaluated in couple with the ultrasonic treatment [124]. None of the tested solvents showed a peel-off efficiency higher than 10% under the optimized conditions (60 °C) but with no sonication. With the addition of the sonication, the authors observed the peel-off efficiency raised by more than six times and equaled 99% for the most effective NMP solvent. On the other hand, Song et al. reported DMAC to possess the best peel-off and cost-efficiency in the row of DMF-DMAC-NMP [125]. Other reagents to remove the polymeric binder from the cathode material are molten salts, particularly, AlCl$_3$-NaCl, which promote the melting of the PVdF binder. For example, Wang et al. achieved an excellent performance at 160 °C using the 1:10 g mL$^{-1}$ ratio of the cathode to molten salt [118]. The situation changes, when, instead of PVdF, polytetrafluoroethylene (PTFE) is a binder—the latter cannot be dissolved in NMP due to its non-polarized nature. To remove PTFE, one of the options is to use trifluoroacetic acid (TFA) in treating the Li-NMC cathodes [126]. The 15 vol.% TFA water solution, 1:8 g mL$^{-1}$ ratio, and 40 °C are enough for detaching the active material. However, TFA possesses a quite low acidity, so Ni, Mn, and Co, tend to get dissolved along with Al.

Another method to peel off the active material is to dissolve an Al substrate in bases. Beforehand, removing Al gets the leachate solution rid of Al$^{3+}$ ions that interfere quantitative precipitation of Co compounds. As an example, 10 wt.% NaOH can be utilized to initiate the dissolution reaction of 1:10 g mL$^{-1}$ Al:NaOH mixture at 25 °C [127]. The concentrations of the target metals dissolved in NaOH were negligibly low herewith. The pretreatment with NaOH is also attractive due to the neutralization of hazardous LiPF$_6$; the contact between LiPF$_6$ and even atmospheric water may generate the HF gas. One should not forget that the treatment with NaOH does not remove the binder, so, after Al dissolution, either solvent (NMP, DMAC, DMF, etc.) or thermal treatment should be implemented. Despite the promising features of described solvents, the problems of affordability, PVdF-solvent separation, and solvent reuse should be solved prior to the industrial implementation.

Leaching the Active Cathode Material

After removing a polymeric binder and utilizing Al foil, the filtered, washed, and dried residue consists of the active cathode material (LCO, NCA, NMC, etc.) and conductive carbon-contained particles. The next step comprises the dissolution of cathode materials by the special leaching compositions transferring the metals as ions to the liquid phase. Depending on the leachate composition, the process can be categorized into acid leaching (inorganic or organic), base leaching, bioleaching, etc. Moreover, especially in the case of acid leaching, the process can go with the addition of a reducing agent—it promotes the leaching efficiency and suppresses the emission of harmful gases.

The inorganic leaching with no reducing agent was the first approach to dissolve metals from the cathode. In 1998, Zhang et al. showed hydrochloric acid to be the most efficient leaching agent among the tested [128]. The main issue was the emission of Cl$_2$ which requires an additional downstream process to capture the gas in the plant

environment. Complimentary to HCl, such acids as $HNO_3$, $H_3PO_4$, and $H_2SO_4$ were studied intensively but appeared to be less effective than HCl due to the lower acidity, corrosiveness, and reducibility [129,130]. The problems of emitted gaseous pollutants and hard purification took place as well [120,131]. Coupling an acid with a reducing agent allowed a lower concentration of the former with no loss of efficiency [132], suppressing the formation of harmful gases, and simplifying the Co dissolving [133]. For example, Meshram et al. showed that when no reducing agent was applied, the leaching efficiencies of Co and Mn metals did not exceed 67%, because they were also present in the solution in their high oxidation states [134]. However, after the addition of the reductor, the efficiencies have jumped to more than 95% for both metals. Zhuang et al. proposed the mixture of phosphoric and citric acids (0.6 M in total) as a leaching agent due to its lower acidity and corrosivity [135]. The optimized leaching conditions, 90 °C, 0.5 h, and the liquid/solid ratio of 50 mL/g, resulted in a leaching efficiency of more than 91%. Note that here citric acid was used as both a reducing and leaching agent. Vieceli et al. investigated $H_2O_2$, $Na_2SO_3$, and $Na_2S_2O_5$ as the reducing agents of the leaching solution based on $H_2SO_4$ [136]. It was shown that $Na_2S_2O_5$ provided the highest leaching efficiency of Ni, Mn, and Co.

Organic acid leaching has attracted much attention as an alternative to inorganic due to its lower corrosivity, acidity, degradability, and toxicity [137]. The inorganic leaching may produce harmful gases such as $SO_x$ or $NO_x$ that require expensive and massive neutralizing and capturing devices [138]. Organic leaching has fewer such problems, so it is considered a more environmentally benign technique. A slightly reducing nature along with the ability to chelate transition metal ions enhances a potential interest in the organic leaching process [139]. For example, promising and widely-investigated citric acid possesses three carboxylic acid groups contributing to solution acidity and forming stable metal chelate complexes. Shih et al. compared the leaching efficiencies of 1.25 M citric with 2 M sulfuric acids with and without the reductant [140]. With no $H_2O_2$ used, they achieved the efficiency of 29 and 75% for sulfuric and citric acids accordingly, whereas it approaches 100% with the reductant. It is seen that by using the leaching with no reductant and keeping the same pH, organic acids can significantly outperform their inorganic counterparts. Oxalic acid, despite promising acidity ($pKa_1 = 1.23$ and $pKa_2 = 4.19$), tends to precipitate the transition metals in the form of insoluble oxalates that limits the leaching efficiency [141]. However, it can be still used for selective leaching of $Li^+$ ions [142] or one-step $CoC_2O_4$ recovery from the LCO cathodes [143]. It is worth mentioning that the soft organic acids are quite tolerant to Al current collectors, which eliminates the stage of separating the cathode material from the Al foil [144]. Li et al. have complemented this research and studied the NMC's metal coordination after acetic and maleic acid leaching [145]. Based on the asymmetric and symmetric stretching vibrations, they concluded that acetic acid provides both monodentate and bridging coordination, whereas maleic acid initiates only bridging. The latter is more stable and does not lead to segregation or impurities. Other investigated organic leaching agents comprise aspartic acid [146], glycine [147], and malic acid [148]. Besides $H_2O_2$, organic-based, green reductants might be used in the organic leaching process such as glucose, sucrose, or cellulose [148]. For instance, Chen et al. illustrated D-glucose to be oxidized during leaching to $CO_2$, $H_2O$, and several organic acids that, in their turn, facilitate the NMC dissolving process [149]. What is more, Pant et al. showed that fruit citrus juice, consisting among other things of ascorbic, malic, and citric acid, can serve as both a reducing and leaching reagent [150]. They managed to achieve leaching efficiencies of more than 94% for all the metals present in the Li-NMC cathode. As we have realized, organic acids are more eco-friendly and no less efficient than inorganic ones in a laboratory environment. However, more industrially oriented investigations should be implemented considering the relatively high price of the organic acids and the partially studied mechanism of leaching.

Alkaline systems are now considered promising leaching agents for LIBs treatment. Ammoniacal solutions are found to be more environmentally friendly and specific but less efficient than acids, so alkaline treatment requires higher leaching times and concentrations. In such alkaline systems, ammonia in the buffer ($NH_3/NH_4^+$, pH = 8–10.5) serves as a leaching agent, whereas different sulfur-containing species are used as reducing agents. As a rule, ammoniacal leaching is selective to $Li^+$, $Ni^{2+}$, $Co^{2+}$, and $Cu^{2+}$ leaving other metal ions ($Al^{3+}$, $Fe^{3+}$, and $Mn^{2+}$) deposited as hydroxides [151]. The nature of this method is a chelating ability of $NH_3$ towards the first line of metals above: ammonia tends to dissolve Ni and Co more easily than Mn. For Ni and Co, the $[M(NH_3)_n]^{2+}$ complexes are stable in the pH range of 8.5–11, whereas Mn-contained ammonia complexes are decomposed to oxides or carbonates [152]. Zheng et al. carried out the alkaline leaching with the optimum solution composition of 4 M $NH_3$-1.5 M $(NH_4)_2SO_4$ + 0.5 M $Na_2SO_3$ [153]. They experimentally confirmed that the sulfite-reducing agent significantly improves the leaching efficiency resulting in a more than 98% recycling rate for Li, Ni, and Co and only 4% for Mn. Another study proposed to perform a two-step leaching process, i.e., to leach again the black solid residue left after the first iteration [154]. The necessity is ascribed to the slow leaching kinetics and dictated by the precipitation of $(NH_4)_2Mn(SO_3)_2$ covering the active material particles. As an alternative approach, Wang et al. proposed adding graphite to the cathode initiating a thermo-mechanochemical reaction before leaching [155]. It allowed to avoid the manganese sulfite precipitate and demonstrated 81% leaching efficiency for Li and more than 96% for Ni and Co. Overall, the ammoniacal LIBs cathode leaching appeared to be a promising alternative for the more developed acid leaching due to its specificity (avoids Mn dissolution) and eco-friendliness (ammonia can be reused). Still, the main limitation of ammoniacal leaching is the high stability of metal–ammonia complexes that complicates separation, solvent extraction, precipitation, and recovery of the target valuable metals.

In recent years, a bio-metallurgical method has become widely investigated for LIB cathode leaching. The metabolites, excreted from bacteria and fungi and containing organic acids, extract the metals from the LIB cathode waste as conventional hydrometallurgy does. One can divide the bioleaching into three types [156]: (i) redoxolysis (involving redox biochemical reactions); (ii) acidolysis (the protons of bio-produced acids act); and (iii) complexolysis (complexes formation stimulates metals dissolution). Switching to the exact examples, Bahaloo-Horch et al. implemented the secretion of *Aspergillus niger* contained oxalic, malic, and citric acids towards the EoL LIB cathodes [157]. The disadvantage lay in the long, 2-week fungus incubation period and the high liquid-solid ratio of the leaching system not suitable for a large scale. Many studies are based on *Acidithiobacillus ferrooxidans* which produce sulfuric acid and ferrous iron. Mishra et al. investigated the leaching assisted by *Acidithiobacillus* during 20 days at pH = 2.5, 30 °C, and a liquid/solid ratio of 10 [158]. They reported 10 and 55% efficiencies of Li and Co correspondingly. Additionally, Niu et al. managed to achieve 89 and 72% efficiency towards Li and Co using the liquid:solid ratio of 50 [159]. *Leptospirillum ferriphilum* and *Acidithiobacillus* thiooxidans were studied by Xin et al. and applied to several types of LIB cathodes including LMO and NMC [160]. The former species consume sulfur to produce $H_2SO_4$, whereas the latter extract ferrous iron from pyrite to reduce $Mn^{4+}$ and $Co^{3+}$. When the liquid:solid ratio equaled 100, the Li, Ni, Mn, and Co extraction level approached 99% after 9 days of exposure in comparison with the 90 min needed for $H_2SO_4$ to achieve the same result [161]. The hybrid system was proposed by Dolker and Pant and comprised a mixture of *Lysinibacillus* species and citric acid [162]. Co was firstly leached out of the cathodes and then absorbed by *Lysinibacillus* contained mixture with an efficiency of 98%. In other works, $Cu^{2+}$ (0.75 g/L) was used to catalyze the bioleaching process to take up to 6 days with the Co recover efficiency of ~100% [163]. The reported addition of $Ag^+$ also increased the Co leaching efficiency to 98% and decrease the leaching time to 7 days [164]. Overall, bioleaching can be potentially promising for industrial LIBs recycling due to its environmental neutrality and low cost. However, continuous preparation time (culturing a microorganism in solutions of

heavy metal ions), slow leaching kinetics (dissolution takes several weeks), and inefficient Co separation (low-purity products' grade), significantly inhibit scaling the bio-metallurgy. All the described leaching technologies are summarized in Table 1.

**Table 1.** Summary of the most efficient reported hydrometallurgical processes of LIBs.

| Cathode Material | Leaching Solution Configuration | T, °C | Time, h | Efficiency, % | | | | Ref. |
|---|---|---|---|---|---|---|---|---|
| | | | | Li | Co | Ni | Mn | |
| **Inorganic acid leaching** | | | | | | | | |
| $LiCoO_2$ | 4 M HCl; Solid:Liquid ratio, g/L (S/L) = 10 | 80 | 1 | 99 | 99 | - | - | [128] |
| $LiCoO_2$ | 1 M $HNO_3$ + 1.7 vol.% $H_2O_2$; S/L = 20 | 75 | 1 | 95 | 95 | - | - | [165] |
| $LiNi_{0.1}Mn_{0.1}Co_{0.3}O_2$ | 1 M $H_2SO_4$; S/L = 50 | 95 | 6 | 93 | 66 | 96 | 50 | [134] |
| $LiNi_{0.5}Mn_{0.3}Co_{0.2}O_2$ | 0.2 M $H_3PO_4$ + 0.4 M citric acid; S/L = 20 | 90 | 0.5 | 100 | 92 | 93 | 92 | [135] |
| $LiNi_xMn_yCo_zO_2$ | 1.25 M $H_2SO_4$ + 0.1 M $Na_2S_2O_5$; S/L = 100 | 60 | 0.5 | 96 | 94 | 89 | 85 | [136] |
| $LiNi_xMn_yCo_zO_2$ | 2 M $H_2SO_4$ + 1 vol.% $H_2O_2$; S/L = 25 | 90 | 1 | 98 | 100 | 88 | 89 | [140] |
| $LiNi_xMn_yCo_zO_2$ | 2 M $H_2SO_4$ + 4 vol.% $H_2O_2$; S/L = 50 | 50 | 2 | 99 | 98 | 98 | 98 | [166] |
| $LiNi_xMn_yCo_zO_2$ | 1 M $H_2SO_4$ + 0.075 M $NaHSO_3$; S/L = 20 | 95 | 4 | 97 | 92 | 96 | 88 | [167] |
| **Organic acid leaching** | | | | | | | | |
| $LiCoO_2$ | 1 M maleic acid + 0.02 M ascorbic acid; S/L = 2 | 80 | 6 | 100 | 97 | - | - | [133] |
| $LiCoO_2$ | 1.25 M citric acid + 1 vol.% $H_2O_2$; S/L = 20 | 90 | 0.5 | 100 | 90 | - | - | [138] |
| $LiCoO_2$ | 1 M oxalic acid; S/L = 50 | 80 | 2 | 99 | 98 | - | - | [143] |
| $LiNi_xMn_yCo_zO_2$ | 2 M formic acid + 6 vol% $H_2O_2$; S/L = 50 | 60 | 0.5 | 98 | 100 | 100 | 100 | [144] |
| $LiCoO_2$ | 1.5 M DL-malic acid + 2 vol% $H_2O_2$; S/L = 20 | 90 | 0.5 | 100 | 90 | - | - | [146] |
| $LiCoO_2$ | 0.5 M glycine + 0.02 M ascorbic; S/L = 2 | 80 | 6 | 95 | 95 | - | - | [147] |
| $LiNi_xMn_yCo_zO_2$ | Citrus juice; S/L = 50 | 90 | 0.3 | 100 | 94 | 98 | 99 | [150] |
| **Alkaline leaching** | | | | | | | | |
| $LiNi_xMn_yCo_zO_2$ | 4 M $NH_3$ + 1.5 M $(NH_4)_2SO_4$ + 0.5 M $Na_2SO_3$; S/L = 10 | 80 | 5 | 95 | 81 | 90 | 4 | [153] |
| $LiNi_xMn_yCo_zO_2$ | 4 M $NH_3$ + 1.5 M $NH_4Cl$ + 0.5 M $Na_2SO_3$; S/L = 10 | 80 | 5 | 93 | 98 | 98 | <10 | [154] |
| $LiNi_{0.5}Mn_{0.3}Co_{0.2}O_2$ | 4 M $NH_3$ + 1.5 M $NH_4HCO_3$ + 30 wt.% $H_2O_2$; S/L = 20 | 60 | 8 | 76 | 95 | 96 | - | [155] |
| **Bioleaching** | | | | | | | | |
| $LiNi_xMn_yCo_zO_2$ | *Aspergillus niger* (produces citric, gluconic, malic, oxalic acids) | 30 | 336 | 100 | 64 | 54 | 77 | [157] |
| $LiCoO_2$ | *Acidithiobacillus ferrooxidans* ($H_2SO_4$, $Fe^{3+}$) | 30 | 480 | 10 | 55 | - | - | [158] |
| $LiCoO_2$ | *Alicyclobacillus + Sulfobacillus*; S/L = 20 | 30 | 720 | 89 | 72 | - | - | [159] |
| $LiNi_xMn_yCo_zO_2$ | *Acidithiobacillus + Leptospirillum*; S/L = 20 | 30 | 168 | 95 | 96 | 96 | 95 | [160] |
| $LiCoO_2$ | *Lysinibacillus* + 0.05 M citric acid; S/L = 5 | 37 | 720 | 25 | 98 | - | - | [162] |

Separation of Valuable Metals

　　　As soon as we dissolved the cathode material in the leaching solution, the next task was to separate the metals from each other. There are several separation solutions currently developing as lab-scale and industrial approaches: solvent extraction and selective salt precipitation from the leaching solution. The solvent extraction implies the distribution of metals through the aqueous and non-aqueous phases. The extraction solution contains an active component dissolved in an organic nonpolar solvent such as kerosene or toluene. Traditionally, the extraction chemistry dissolved in an organic solvent is brought to contact with the mother solution, intensively mixed, and then separated by the difference in densities. Then, the scrubbing step is used to remove co-extracted contaminants. In the end, the target metal gets extracted to the aqueous phase by the stripping reaction, the reverse extraction process [168]. The solvent extraction technique was adopted from the industrial method of separation of Ni and Co [169]. For example, bis(2,4,4-trimethylpentyl) phosphonic acid (Cyanex 272, Conquest 290) is used in laterite processing in sulfate media, whereas tri-isooctyl amine is for chloride media. It illustrates that, for a certain leaching acid, the extracting agent should be chosen accordingly. Depending on the conditions (leachate media, pH, temperature, organic/aqueous ratio), the extraction selectivity varies from ion to ion. Particularly, Cyanex 272 at the sulfate media fully extracts Ni at pHs above 6, Mn and Co at 5, and Al at 3 [170,171]. Takahashi et al. have investigated the Co extraction from the LCO leachate by Cyanex 272 [161,172]. They demonstrated a high-performance separation of Co and Li throughout the phases. Tsakiridis et al. managed to obtain a high-pure Al solution with a negligible amount of Ni and Co by the mixture of 20% Cyanex 272 + 5% tributyl phosphate (TBP) at 40 °C and pH = 3.0 [153]. The Cyanex 272 chemistry was applied by Ichlas et al. to separate ions leached from NMC cathodes in the nitrate media [173]. In the beginning, 20% Cyanex 272 extracts Al and partially Co followed by scrubbing Co traces to the aqueous phase. Then, it was again put in touch with Cyanex 272 to withdraw Co and Mn cations. Nayl et al. applied Cyanex 272 at a certain range of pH values to consistently extract the valuable metals from the mother solution. Varying pH from 3.5 to



8, they achieved a recovery rate of metals higher than 89% [174]. Swain et al. demonstrated Cyanex 272 with 5% TBP in kerosene to provide more than 85% of Co separation at pH = 5 [175]. Alternatively, Jha et al. achieved 99.9% Co extraction efficiency at pH = 5 with isodecanol as a phase modifier [176]. Dorella et al. after the cathode material dissolving in $H_2SO_4 + H_2O_2$ utilized 0.72 M Cyanex 272 at 50 °C—achieving the Co extraction efficiency of 85% [177]. Bis(2-etylhexyl) phosphoric acid (D2EHPA) is another extractant tested in the recycling of LIBs. Wang et al. separate Co with 99.5% efficiency at pH = 2.2–2.7 using the combination of D2EHPA and bis(2-ethylhexyl)-2-ethylhexylphosphonate (PC-88A) from the sulfuric-based leachate solution [178]. At the same time, D2EHPA was proved to be more selective to $Mn^{2+}$ capturing ~10% of Co along with [179,180]. To compare anion impact on the extraction efficiency, Dhiman and Gupta utilized Cyphos IL 102 and $H_2SO_4$, HCl, and $HNO_3$-based leachate solutions [181]. The authors found the extraction efficiency in the case of HCl to increase with the rise of concentration, whereas the extraction of the metal from $H_2SO_4$ and $HNO_3$-containing sample was unchangeably low. These results support the thesis that the choice of a certain extractant and extraction conditions depends on the chemistry used in recycling. Some attempts were done with the leachate solution formed by citric acid. D2EHPA was particularly shown to selectively extract $Mn^{2+}$ from the media due to the strong chelating interaction between $Co^{2+}$ and citrate anions [182]. Despite the high selectivity and ~100% separated solution purity, a broad industrial implementation of the solvent extraction method is limited by the high price of reagents and equipment complexity. However, some processes (e.g., JX Nippon [183]) already use the discussed methodology. Particularly NMC LIBs, which after the dissolution in sulfuric acid are treated with D2EHPA to sort Mn out. After that, PC-88A is used to consistently separate Co and Ni at pH of 4.2 and 6.5, respectively.

Another class of chemicals that might be used for selective ion separation is ion-exchange resins. In this case, a solid resin is added to the leachate providing a reversible ion exchange through the solid–liquid interface. The essence of the method lies in the resin's functional groups selective towards certain ions. The interaction is conditioned by either electrostatics or chelating coordination with the target metal ion [184]. The commonly used functional groups are iminodiacetate (Purolite S930, Lewatit TP 207) and bis-picolylamine group (Lewatit TP 220, Dowex M4195) [185–188]. Each type requires tuning pH to extract specific target ions, so experiments should be planned accordingly [189]. As an example, Strauss et al. applied Dowex M4195 to NMC cathodes and achieved 99 and 98% efficiency for Ni and Co, respectively [190]. Another perspective type of resin contains aminophosphonate functional groups (Lewatit TP 260, Purolite S950) and allows to sort Mn, Al, Cu, and Fe ions [191]. The ion-exchange resins are more beneficial than solvent extraction in terms of the smaller amount of material and lower cost. At the same time, the separation rate of the metals is lower, so the combination of the two is considered to achieve the best performance and economic value.

The second approach to separate the target metal ions is selective precipitation from the leachate solution. The simplest method is to switch pH and thus consistently deposit metals as hydroxides. In order to increase pH, one can use either a concentrated alkali reagent or a saturated ammonia solution. Obviously, $NH_3 \cdot H_2O$ is a softer base than NaOH, so it causes more dilution than the latter. Moreover, ammonia may coordinate metal cations forming soluble $[M(NH_3)_6]^{2+}$ complexes that complicate the deposition. Except for pH, the deposition temperature plays a crucial role in efficient salt precipitation. $CoC_2O_4$ formation is endothermic, whereas some hydroxide and carbonate deposition reactions are endothermic [192–194]. The other minor factors influencing salt precipitation are interionic interactions and the formation of exotic insoluble species [195,196]. The main pitfall of the pH-based deposition is an undesired coprecipitation (between Cu, Ni, and Co), which is quite hard to predict. At the same time, Suzuki et al. dissolved $Cu^{2+}$, $Al^{3+}$, and $Co^{2+}$ of equimolar concentration in $H_2SO_4/H_2O_2$ [197]. At pH = 7, the authors observed nearly full deposition of $Al(OH)_3$ and $Cu(OH)_2$ and 50% of $Co(OH)_2$ as a coprecipitate. At low concentrations, $Cu^{2+}$ can be precipitated with high selectivity, while on the equimolar basis,

selectivity of Cu separation is unlikely possible. Kang et al. proposed a modified approach to avoid such losses of valuable metals [198]. At pH = 6.5 and using NaOH and $CaCO_3$, they first precipitated all the metals at once and then washed this precipitate to return the valuable metals into the leachate solution. This approach allowed a 99% efficient removal of the impurity ions yet provided some losses of the target materials. Another approach is to oxidize $Co^{2+}$ to $Co^{3+}$, since the pH overlap between $Ni(OH)_2$ and $Co(OH)_3$ is much more narrow than between those of $Ni^{2+}$ and $Co^{2+}$. Joulié et al. experimentally implemented this idea with NaClO as an oxidant [128]. The authors recycled NCA cathodes and achieved ~100% Ni and Co recovery efficiency. Some reports addressed the carbonate precipitation under the pH control as more selective than the hydroxide. At pH = 7.5, $MnCO_3$ was successfully precipitated with 92% efficiency, whereas at pH of 9, $NiCO_3$ was formed with an 89% yield [167]. Nayl et al. confirmed the values above reporting 94% efficiency of $MnCO_3$ precipitation at pH = 7.5 and 91% efficiency of $NiCO_3$ at pH = 9 [199]. Taking into account the issue of Cu co-precipitation and the close solubilities of Ni, Mn, and Co hydroxides and carbonates, we are expected to use other precipitation agents to selectively separate valuable metals from the others. Cai et al. developed the method of using $Na_2S$ to separate Mn and Co in the form of sulfides [200]. The selective dissolution of sulfides was performed by the diluted acetic acid: at pH = 4.74, almost all MnS was dissolved with CoS remaining in the solution. He et al., after sorting $Mn^{2+}$ out, recovered CoS via $(NH_4)_2S$ adding to the solution [201]. Choubey et al. repeated the procedure applying to the NMC cathodes and successfully precipitated CoS at pH = 3 and 25 °C with a negligible amount of Li, Mn, and Ni (<1%) [202].

A popular complexation agent used for $Ni^{2+}$ in basic conditions is dimethylglyoxime (DMG). The advantage of DMG is the relative ease of recovering after the separation—the $Ni(DMG)_2$ complex can be simply dissolved by HCl. Chen and Zhou estimated optimum conditions for Ni precipitation from the citric leachate to be $DMG/Ni^{2+}$ = 2.95 and pH = 8 with insignificant precipitation of other metals <1% [182]. In sulfuric acid, $DMG/Ni^{2+}$ = 2, and pH = 5, $Ni^{2+}$ can be also recovered with close to 100% purity and selectivity [166]. Ammonia solution can be used for $Ni(DMG)_2$ precipitation via an intermediate formation of the $[Ni(NH_3)_6]^{2+}$ complex [203]. Oxalate anions $(C_2O_4^{2-})$ are often reported to be selective to $Co^{2+}$ at a pH of 1.0–1.5 [196], so $C_2O_4^{2-}$-containing agents are the best choice for LCO cathodes. For example, Meshram et al. precipitated $CoC_2O_4$ at 50 °C and pH = 1.5 and achieved ~96 and 99% efficiency and selectivity, respectively [167]. $CoC_2O_4$ can be easily decomposed to $Co_3O_4$ which serves as a precursor for cathode resynthesis or catalyst preparation. For $Mn^{2+}$ selective precipitation, potassium permanganate can be used to obtain insoluble oxide species of manganese in higher oxidation states, +3 and +4 [204]. Being a $H^+$ concentration-dependent reaction, $Mn^{2+}$ selective oxidation by $KMnO_4$ is sensitive towards pH of the media; at a pH higher than 3, Ni and Co precipitation may be observed. Ammonium persulfate, another oxidizer used for $Mn^{2+}$, as well as $KMnO_4$ requires fine-tuning of the molar ratio and pH to avoid $Co(OH)_2$ deposition [205]. The only limitations of $(NH_4)_2S_2O_8$ are the cost and $Cl^-$ oxidation to gaseous $Cl_2$ if the leachate solution contains HCl. The precipitation of lithium salts is usually performed by the carbonate- or phosphate-containing chemicals (sodium salts or acids) obtaining $Li_2CO_3$ or $Li_3PO_4$. Precipitation is carried out at pH > 11 after solution concentrating or at 100 °C due to the low solubility of $Li_2CO_3$ [194]. Because of the high precipitation efficiency and purity (>99.9%), $Li_2CO_3$ can be directly used in the second cycle cathode resynthesis.

The described methods of the metals separation technologies are summarized in Table 2.

**Table 2.** Approaches of valuable metals separation and recovery.

| Ions Presented in a Leachate | Separation/Recovery Methodology | Extraction, % | | | | Ref. |
|---|---|---|---|---|---|---|
| | | Li | Co | Ni | Mn | |
| | **Solvent extraction** | | | | | |
| $Li^+$, $Co^{2+}$, $SO_4^{2-}$ | (1) Co extraction by Cyanex 272 at organic/aqueous ration (O/A) = 1, 25 °C, pH = 4. 4 times repeating; (2) Co stripping with 2 M $H_2SO_4$ at pH = 0–1. 4 times repeating; | - | 85 | - | - | [130] |
| $Fe^{3+}$, $Cu^{2+}$, $Al^{3+}$, $Li^+$, $Ni^{2+}$, $Mn^{2+}$, $Co^{2+}$, $SO_4^{2-}$, $NH_4^+$ | (1) $Fe^{3+}$, $Cu^{2+}$, $Al^{3+}$ impurities extraction with 20% Acorga M5640 in kerosene at pH = 1.0–2.2; (2) $Li^+$, $Ni^{2+}$, $Mn^{2+}$, $Co^{2+}$ extraction by 0.04 M Na-Cyanex 272 in kerosene at O/A = 1, pH = 5, 25 °C. Scrub Ni and Li with $Na_2CO_3$ at pH = 9.0 and 12.0; (3) Strip Mn and Co with 0.1 M $H_2SO_4$ at O/A = 0.5 and precipitate $MnCO_3$ at pH = 7.5 and $Co(OH)_2$ at pH = 11.0; | 99.6 | 99.0 | 99.4 | 99.7 | [174] |
| $Cu^{2+}$, $Al^{3+}$, $Li^+$, $Ni^{2+}$, $Mn^{2+}$, $Co^{2+}$, $SO_4^{2}$ | (1) D2EHPA in kerosene to extract Cu and Mn at O/A = 1, pH = 2.6–2.7, 25 °C; (2) PC-88A to separate Ni and Co at O/A = 1, pH = 4.25, 25 °C; (3) Stripping Co with 0.5 M oxalate; | - | 80 | - | - | [178] |
| $Cu^{2+}$, $Al^{3+}$, $Li^+$, $Ni^{2+}$, $Mn^{2+}$, $Co^{2+}$, $Cl^-$ | (1) $Fe(OH)_3$ precipitation by NaOH at 95 °C and pH = 3.0–3.5; (2) $MnO_2$ precipitation by 10% $(NH_4)_2S_2O_8$ at 70 °C and pH = 4; (3) $Cu(OH)_2$ and $Al(OH)_3$ precipitation at pH = 5.5; (4) 0.2 M Cyphos IL 102 extracted Co, and $CoC_2O_4$ was precipitated; (5) Ni separated from Li by $NH_{3(aq.)}$ and DMG at pH = 9; (6) $Li_2CO_3$ deposited by NaOH + $Na_2CO_3$ at 100 °C and pH = 11; | 99.6 | 98.6 | - | 99.9 | [181] |
| $Li^+$, $Ni^{2+}$, $Mn^{2+}$, $Co^{2+}$, citric | (1) Ni precipitated with 0.05 M DMG at pH = 6 and recycled as $NiCl_2$ by further 1 M HCl treatment; (2) Co precipitated by $(NH_4)_2C_2O_4$ at pH = 6 and 55 °C; (3) Mn separated by 20 vol.% D2EHPA at O/A = 2, pH = 6; (4) Li recovered as $Li_3PO_4$ by 0.5 M $Na_3PO_4$; | 89 | 97 | 98 | 97 | [182] |
| | **Extraction by ion-exchange resins** | | | | | |
| $Ni^{2+}$, $Mn^{2+}$, $Co^{2+}$, $Mg^{2+}$, $SO_4^{2-}$ | Amberlite IRC 748 (carries iminodiacetate functional groups) at pH = 4–5 selectively extracts Ni and Co via automated titration column set-up; | - | 100 | 100 | - | [185] |
| $Fe^{3+}$, $Al^{3+}$, $Mn^{2+}$, $Ni^{2+}$, $Co^{2+}$, $Mg^{2+}$, $SO_4^{2-}$ | (1) Lewatit TP 220 separated Ni and Co; (2) 6.8 wt.% $NH_{3(aq.)}$ solution eluted Ni and Co; | - | 85 | 95 | - | [188] |
| $Fe^{3+}$, $Al^{3+}$, $Zn^{2+}$, $Li^+$, $Ni^{2+}$, $Mn^{2+}$, $Co^{2+}$ | Using two-column set-up Dowex M4195 separated Ni at pH = 1.1 and Co at pH = 4.1; | - | 98.5 | 99.0 | - | [190] |
| $Fe^{3+}$, $Al^{3+}$, $Cu^{2+}$, $Li^+$, $Ni^{2+}$, $Mn^{2+}$, $Co^{2+}$ | (1) Lewatit TP260 (contains aminomethylphosphonic groups) separated Li, Co, and Ni of battery grade purity; (2) Cu and Mn were eluted with 2 M $H_2SO_4$, Fe and Al—with 0.4 M $K_2C_2O_4$; | 99.6 | 99.6 | 99.6 | - | [191] |
| | **Selective precipitation** | | | | | |
| $Fe^{3+}$, $Al^{3+}$, $Cu^{2+}$, $Li^+$, $Co^{2+}$, $Cl^-$ | Subsequent addition of 40% NaOH until pH = 6 to remove impurities (Fe, Al, Cu); | - | 89 | - | - | [112] |
| $Fe^{3+}$, $Cu^{2+}$, $Li^+$, $Mn^{2+}$ $Co^{2+}$, $SO_4^{2-}$ | (1) Removing Fe as sodium jarosite by 10 wt.% $Na_2SO_4$ at pH = 3, 95 °C for 2 h; (2) $MnO_2$ deposition by $(NH_4)_2S_2O_8$ at pH = 4 and 70 °C; (3) $Cu(OH)_2$ precipitated at pH = 5.5; (4) Co extraction from Li and Ni with 25 wt.% P507 in kerosene; (5) $CoC_2O_4$ precipitation by $(NH_4)_2C_2O_4$ at pH = 1.5; | 97 | 98 | 97 | 99 | [196] |
| $Fe^{3+}$, $Al^{3+}$, $Cu^{2+}$, $Li^+$, $Ni^{2+}$, $Mn^{2+}$, $Co^{2+}$ | (1) Removing Fe, Al, and Cu by adding 4 M NaOH and 50 wt.% $CaCO_3$ solution at pH = 6.5; (2) Extraction of Co with 0.5 M Cyanex 272 in kerosene at pH = 6.0, O/A = 2; (3) Stripping Co with 2 M $H_2SO_4$; | - | 92 | - | - | [198] |
| $Al^{3+}$, $Li^+$, $Ni^{2+}$, $Co^{2+}$ | (1) NaClO added to transit $Co^{2+}$ to $Co^{3+}$ ($Co_2O_3$), $ClO^-/Co^{2+}$ = 3, at pH = 3; (2) $Ni(OH)_2$ deposited at pH = 11; (3) $Ni(OH)_2$-contained precipitate washed with base to solubilize Al into $Al(OH)_4^-$; | - | 90 | 96 | - | [129] |
| $Li^+$, $Ni^{2+}$, $Mn^{2+}$, $Co^{2+}$, $SO_4^{2-}$ | (1) $MnCO_3$ precipitation by saturated $Na_2CO_3$ at pH = 7.5 adjusted by 2.0 M NaOH; (2) $NiCO_3$ precipitation by stirring for 1 h at 25 °C; (3) $Co(OH)_2$ by saturated NaOH for 2 h at pH = 11–12; (4) $Li_2CO_3$ by $Na_2CO_3$ at pH = 12 for 1 h stirring; | 90 | 95 | 91 | 94 | [199] |
| $Li^+$, $Mn^{2+}$, $Co^{2+}$, $SO_4^{2-}$ | (1) 1 M $Na_2S$ added to precipitate MnS and CoS; (2) MnS selective dissolution by proper amount of 0.05 M acetic acid; (3) Separation; $Mn(OH)_2$ precipitation by 2 M NaOH; (4) $Li_3PO_4$ precipitation by 1 M $Na_3PO_4$; | - | 99 | - | 98 | [200] |
| $Li^+$, $Ni^{2+}$, $Mn^{2+}$, $Co^{2+}$, $Cl^-$ | (1) $MnO_2$ recovery by $KMnO_4$ ($Mn^{2+}/Mn^{7+}$ = 2) at pH = 2 and 40 °C; (2) Saturated $NH_{3(aq.)}$ solution is added to the leachate; (3) $Ni(DMG)_2$ precipitation (DMG/$[Ni(NH_3)_6]^{2+}$ = 2) at pH = 9 and 25 °C; then, can be recovered as $Ni(OH)_2$ by 1 M NaOH at pH = 11; (4) $Co(OH)_3$ recovery from $[Co(NH_3)_6]^{3+}$ by 1 M NaOH at pH = 11; (5) $Li_2CO_3$ precipitation by $Na_2CO_3$ at 100 °C; | 80 | 97 | 97 | 98 | [203] |

### 2.2.3. Emerging Recycling Techniques
Mechanochemical Pretreatment

In attempts to achieve better recycling performance, many researchers are focused on creating non-standard, emerging approaches for a certain recovery stage. One of them is mechanochemical pretreatment. The procedure implies the deformation of the cathode active material's crystal structure by co-grinding with reducing agents. The resulting decrease in particle size and subproducts simplifies the leaching process, lowers liquid/solid ratios, and decreases leachate concentrations [179]. It was several times reported a selective Li extraction from the mixed cathodes due to the relative weakness

of the Li-O bonds. Yang et al. utilized ball milling of the cathode material with $Na_2S$ and manage to obtain the mixture of LiOH, $Na_2SO_3$, and $Ni_{0.5}Mn_{0.3}Co_{0.2}(OH)_2$ [206]. Then, Li can be selectively leached with distilled water (where $Ni_{0.5}Mn_{0.3}Co_{0.2}(OH)_2$ is insoluble) and further precipitated as $Li_2CO_3$ with 99.9 wt.% purity. Zhang et al. joined the temperature effect to the mechanochemical process [207–209]. They ball-milled the mixed cathode mass with an addition of 20 wt.% lignite carbon and then calcined at 650 °C for 3 h to obtain a mixture of $Li_2CO_3$, Ni, MnO, and Co. After that, $Li_2CO_3$ was converted into soluble $LiHCO_3$, filtered, and precipitated again. The rest of the metals were leached by $H_2SO_4$ with 96% efficiency and a 3.5 mL g$^{-1}$ liquid/solid ratio. Liu et al. ball-milled NMC cathode material with 10 wt.% carbon black and heated the mixture to 550 °C with 0.5 h holding [210]. Under Ar flow, they obtained metals and oxides of Ni, Mn, Co, and $Li_2CO_3$, which were easily leached with water at 30 mL g$^{-1}$ and 25 °C with 93% Li selectivity. Ni, Mn, and Co were dissolved in 4 M $H_2SO_4$ at 10 mL g$^{-1}$ and 90 °C with efficiencies of more than 99.5%. Guan et al. co-grinded the cathode material with Fe fine powder that reduced $Co^{3+}$ to $Co^{2+}$ and improved leaching efficiencies by 2–4.5 times to 77% Li, 91% Co, and ~100% for Ni and Mn [97]. Saeki et al. applied a different material, PVC, accompanying LCO during ball milling [211]. In the air, the main products of the solid-state reaction were Li and Co chlorides which were subsequently leached with deionized water. Wang et al. tested PVC along with a set of inorganic chlorides and EDTA compounds [210]. The authors found EDTA to demonstrate the best performance providing 99% Li and 98% Co recovery efficiencies.

Electrochemical Method

The electrochemical treatment is another method alternative to traditional hydrometallurgy. It allows for avoiding the addition of other substances into the system. The technology described by Lain in 2001 includes electrolyte extraction, electrode dissolution, and cobalt reduction followed by the lithium release from the solid structure [211]. Myoung et al. extended this research and obtained cobalt hydroxide via potentiostatic treatment of LCO and nitric-acid leachate solution [212]. Freitas et al. also studied the Co electrochemical recovery investigating the pH change in the reaction, the pH effect on precipitation, etc. [213–216].

Ionic Liquids

Ionic liquids (IL—salts with a melting point below 100 °C) are becoming more and more spread due to their solubility properties; they have already found many applications, especially in catalysis [216]. Besides, ILs possess low flammability, low vapor pressure, and strong interactions with various inorganic and organic species. A task-specific ILs should be particularly mentioned, as this class demonstrates specificity to certain chemistries [217,218]. For example, by a proper selection of cation and anion, one can switch between salt's hydrophilicity/hydrophobicity [219]. Morizono et al. observed ~45% Co extraction efficiency in a mixture of histidine-2-ethylhexylamide and 1-hexyl-3-methylimidazolium hexafluorophosphate [220]. This finding has pushed further attempts to separate Co in a hydrometallurgical process of LIBs recycling. The Cyphos IL 101 ionic liquid was proved to demonstrate high separation efficiency within the Li/Ni/Co mixture approaching 70–90% [221]. Zante et al. investigated imidazolium-bis(trifluoromethylsulfonyl)imide ILs in tributyl phosphate (TBP) for processing the NCA cathodes [222]. They achieved decent Li/Co and Li/Ni separation factors of up to 9. In the author's further investigation, they utilized 1-butyl-3-methylimidazoliumbis(tri-fluoromethylsulfonyl) imide, tri-hexyl tetradecylphosphonium chloride ([P66614][Cl]), and N,N,N',N'-tetra(n-octyl) diglycolamide to sort the metals out from the NMC-dissolved leachate [223]. The overall Co extraction efficiency approached 93% in the presence of Ni and Li. Another research was performed by Dhiman and Gupta [180]. The authors added 0.2 M Cyphos IL 102 in toluene to the NMC cathode leachate achieving 99.9% separation efficiency. Othman et al. studied the separation efficiency of NMC metals in the HCl leachate solution by tetraoctylphosphonium

oleate ([P8888][oleate]). They achieved 99 and 89% extraction for Co and Mn, respectively, and 100% purity of the final products [224]. Overall, the use of ILs looks promising for industrial applications, primarily due to their safety. However, the nature of the acting reactions and economics should be more evaluated.

Deep Eutectic Solvents

Deep eutectic solvents (DES) are a class of ILs. The mixture of several components is specified to possess the eutectic temperature, the melting/solidifying point, lower than the separate components have. DES has consisted of a hydrogen bond donor and acceptor, i.e., a mixture of Lewis or Brønsted acid and base [225]. Despite physical and chemical properties being similar to that of traditional ILs, DES are even less flammable, easier to synthesize and handle, more environmentally benign, and cheaper. The commonly-used DES is choline chloride, and carboxylic acids; so-called natural deep eutectic solvents (NADES) include amino acids, sugars, choline derivatives, and other metabolites [121]. Recently, numerous research papers are devoted to DES employment specifically for EoL electronics and LIBs recycling [226–230]. Tran et al. investigated LCO and NMC recycling with a mixture of choline chloride and ethylene glycol at 150 °C [226]. Interestingly, DES allowed to efficiently separate the active material from the other cathode components (Al foils, binder, carbon). Wang et al. confirmed this result—the same DES mixture was able to separate Al foil from the NMC cathode material at 180 °C [228]. Later on, the authors approached 95% Li and Co extraction efficiency after the 12 h reaction obtaining $Co_2O_3$ as the final product [227]. The main disadvantage of using the choline chloride+ethylene glycol mixture as a DES may be in their ability to leach Cu and Al [231], so the prior separation of them is needed. Alternatively, Roldán-Ruiz et al. studied the mixture of *p*-toluenesulfonic acid monohydrate and choline chloride [229]. At 90 °C the Li and Co recovery exceeded 94%; $Co_2O_3$ was further obtained by sequential carbonate and thermal treatment. Chen et al. investigated $H_2C_2O_4$ as an additive to the traditional (choline chloride+ethylene glycol) system [230]. At the relatively low temperature (70 °C), the authors extracted Li and Co with >99% efficiency

Supercritical Fluids

The next material type used for spent LIB metals extraction is supercritical fluids (SFs). A matter at a certain temperature and pressure (called at the critical point) possesses no interface (and difference in properties) between gas and liquid phases. Moreover, by slightly altering the parameters one can tune the solubility of the fluid. An increase in pressure and temperature enhances solubility. Overall, SFs have high diffusivity, high solubility, and low viscosity attractive for the efficient extraction of desired metals. The commonly-used SFs are carbon dioxide [231–233] and water [234]. The supercritical $CO_2$ atmosphere is relatively easy to be achieved—the $CO_2$ critical parameters are 31.1 °C and 79.8 bar [235]. Applying LIBs recycling, SFs can be used within solvent extraction techniques to decrease metal salt solubility. Bertuol et al. studied the leaching of LIBs cathodes by the $H_2SO_4+H_2O_2$ mixture [232]. With the use of $CO_2$, the researchers significantly diminished the leaching time from 60 to 5 min achieving >95% Co recovery efficiency. Fu et al. evaluated the combination of $CO_2$ SF and DMSO to recover the polymer binder and separate the cathode material [233]. At 70 °C and 80 bar, they reached the recovery of 98% efficiency after 13 min. Liu et al. implemented supercritical $H_2O$ (374.1 °C and 22.1 bar) in tandem with PVC to enhance its dichlorination [234]. At a solid/liquid ratio of 16 g $L^{-1}$ and 350 °C they approached 98 and 95% leaching efficiency of Li and Co, respectively. Additionally, to beneficial time reduction of the process, the significant limitations of most SFs are high temperature and pressure, specific atmospheres, and complex equipment that drastically increase the total cost of industrial operations.

### 2.3. Methods of Cathode Direct Restoration

2.3.1. Resynthesis

At some point of view, traditional methods of precipitation, solvent extraction, etc. are not so economically favorable due to the large number of used chemistries, routes, amount of emission, and so on. To solve these problems, direct cathode restoration methods are now investigated. One such method is a resynthesis method [147,236–240] carried out right in the solution via coprecipitation. Before the deposition, some extra amount of Ni, Mn, or Co salts are added to achieve the desired molar ratio of the metals in the recovering cathode. After that, the co-precipitated hydroxides are mixed with an excess of $Li_2CO_3$ and annealed to obtain the regenerated active material [241]. For example, Zou et al. leached the cathode scarp with $H_2SO_4 + H_2O_2$, removed Fe by adjusting pH to 3, and balanced the metals' concentration [239]. After that, they increased pH to 11 to precipitate $Ni_{1/3}Mn_{1/3}Co_{1/3}(OH)_2$ and $Li_2CO_3$. Finally, the authors ground the obtained powders together and sintered the mixture at 900 °C. Sa et al. restored the NMC cathode using co-precipitation in the nitrogen atmosphere [236]. The resynthesized cathode provided 50 cycles and 80% of the initial capacity level at the first cycle. Sencanski et al. coprecipitated $Ni_{1/3}Mn_{1/3}Co_{1/3}(OH)_2$ right from the leachate solution and then synthesize NMC. To improve the metal intermediates' stability (especially, $Mn(OH)_2$), He et al. proposed to precipitate metals in the form of carbonates [148]. At pH = 7.5 and 60 °C, they got spherical particles with a uniform size distribution. Hydroxide precipitation can be also carried out consequently for every metal type. Hu et al. raised the pH of the leachate solution to 11 by NaOH and excess amounts of LiOH [238]. After that, the precipitate obtained was thermally treated at 480 and then at 950 °C. The finally obtained products demonstrated the battery-grade purity.

The sol-gel resynthesis method can be applied in the case of leaching with organic compounds, as they can serve as chelating agents. After adjusting pH and Ni:Mn:Co ratio, a solution is heated to remove water, form the gel, immobilize the metal ions, and decompose the organic chemistries. Such a method allows producing particles with a size smaller than 1 μm and controllable layer structure [237]. Li et al. utilized the citric acid+$H_2O_2$ leachate to get metal recovery efficiencies of higher than 95% [145]. After the water was evaporated, the gel was treated firstly at 450 °C to burn out organics and then calcined at 900 °C to finalize the NMC structure. The capacity of the resynthesized cathode was slightly lower, but the cycling stability went up preliminarily due to its doping with Al left in the solution after initial cathode dissolution. At the same time, one should always keep in mind the ease of scalability of the proposed processes. An important aspect is to resynthesize all the types of cathodes using the same sequence of steps and apparatus. To achieve this, Zheng et al. studied NMC regeneration from the cathodes taken out using four different dissembling lines [242]. Despite different cathode configurations (particle size, porosity, density), the authors reached the uniform particle size distribution and density (10 μm, 2.5 g cm$^{-3}$) suitable for LIBs.

Overall, the electrochemical performance of resynthesized material approaches the initial, but still falls behind and differs from leachate to leachate [63]. Besides, an improper ratio of metals in the resynthesized cathodes, high porosity, instability, and low crystallinity limit the capacity of the restored LIBs [26].

2.3.2. Regeneration

As an alternative to resynthesis from the solution, a direct regeneration of spent cathode materials is now also attracting much interest. The regeneration process implies re-lithiating the degraded active material and removing an excess of binder to prevent unnecessary agglomeration. Additionally, a proper material phase and purity can be restored using a solid-state sintering or hydrothermal approach [243–246]. Kim et al. for the first time tested 5 M LiOH to renovate the EoL LCO cathodes [246]. Without any pretreatment, they restored the cathode via the sequence of dissolution and precipitation carried out at 200 °C for 20 h. The initial capacity was 144 mAh g$^{-1}$, whereas its retention

exceeded 92% after 40 cycles. Zhou et al. reported the particles of spent NMC to contain $Li_2CO_3$, $LiMn_2O_4$, and LiF on their surface [243]. The authors implemented $Li(CH_3COO)$ to saturate the material with Li and annealed the system to remove impurities and side phases and eliminate cracks. The restored NMC cathode possessed decent electrochemical properties: an initial capacity of 147 mAh g$^{-1}$ and 89% retention after 100 cycles at 1 C. Li et al. also successfully utilized LiOH to restore Li presence in the NMC structure and obtained a single phase and uniform cations distribution [245]. Shi et al. developed a several-step method involving both hydrothermal and annealing treatment [244]. They managed to restore the layered structure of the mixed oxide, initial morphology, and high electrochemical performance close to the original one. Moreover, the authors established the $O_2$-rich atmosphere to be highly recommended during the sintering to maintain a uniform cation distribution in the structure and phase purity.

The non-standard methods, including the mechanochemical approach, can also directly regenerate the cathodes. As an example, Meng et al. combined annealing with mechanochemical activation to reduce the cathode's particle size [247]. It allowed to diminish ions' free path and, hence, improved the Li intercalation to the layered structure. The capacity of regenerated cathodes approached 165 mAh g$^{-1}$ (0.2 C) and 80% retention after 100 charge/discharge cycles. Ra et al. used another method, called Etoile-Rebatt technology, to regenerate the LCO material [248]. Utilizing the combination of electrochemical and hydrothermal processes, the authors renovated the cathode using 4 M LiOH solution at <100 °C and achieved 135 mAh g$^{-1}$ capacity with a retention of 96% after 50 cycles. The ultrasound strategy was applied to the LCO cathodes by Zhang et al. [249–252]. The treatment with ultrasound can initiate the cavitation effect, locally increase temperature and pressure, and produce oxidative hydroxyl radicals. All of the mentioned catalyzed the LCO regeneration that was carried out hydrothermally at 80–120 °C, 600–1000 W ultrasonication power, and 6–12 h of the process duration. The results proved the renovated LCO to have good morphological, structural, and electrochemical properties. The cathode material with high crystallinity, dispersion, and layered structure demonstrated the 133 mAh g$^{-1}$ capacity with 98% retention after 50 cycles.

Besides, the direct regeneration method can be applied to the LFP cathodes [27]. Indeed, this is a cheap, environmentally benign, and energy-efficient routine that avoids the use of acids and high temperatures. It makes the regeneration profitable for such cathodes as LFP or LMO that are not economically favorable to be recycled by pyro- or hydrometallurgy. The main pitfall of the direct regeneration is still the quality of the restored cathode material. Their electrochemical performance highly depends on the accuracy and, hence, on the process scale level. High requirements for phase purity and temperature regime appeal to a careful design of the process, especially, in the case of mixed feedstock. All of that stimulates the further investigation of the process with the target to scale it up to an industrial stage.

All the discussed LIB restoration protocols are summarized in Table 3.

**Table 3.** Summary of LIB cathode material direct restoration protocols.

| Cathode Material | Restoration Methodology | Performance of Restored Cathode | Ref. |
|---|---|---|---|
| $LiNi_{0.5}Mn_{0.3}Co_{0.2}O_2$ | **Resynthesis** (1) Adjust the selected ratio of Ni, Mn, and Co; (2) Deposit hydroxides via $NH_3$ and NaOH; (3) Co-grind the product with 3%-excess of $Li_2CO_3$; (4) Calcine at 500 °C for 5 h; then at 750–900 °C for 12 h; | 150 mAh g$^{-1}$ at 0.1 C; 80% capacity retention after 100 cycles at 0.5 C; | [236] |

**Table 3.** *Cont.*

| Cathode Material | Restoration Methodology | Performance of Restored Cathode | Ref. |
|---|---|---|---|
| $LiNi_xMn_yCo_zO_2$ | (1) Adjust Ni:Mn:Co ratio by nitrate precursors and pH to 8 by $NH_{3(aq.)}$; (2) After stirring at 80 °C calcine at 400 °C for 2 h and at 800 °C for 8 h; | 147 mAh g$^{-1}$ at 0.5 C; 94% capacity retention after 100 cycles at 0.5 C; | [237] |
| $LiNi_xMn_yCo_zO_2$ | (1) Remove $Fe^{3+}$ ions at pH = 3; (2) Adjust Ni:Mn:Co ratio by sulfate precursors; (3) Coprecipitate hydroxides at pH = 11; (4) Sintering with $Li_2CO_3$ at 900 °C for 15 h; | 130 mAh g$^{-1}$ at 0.4 C; 82% capacity retention after 50 cycles at 0.4 C; | [239] |
| $LiNi_xMn_yCo_zO_2$ | (1) Adjust Ni:Mn:Co ratio by sulfate precursors; (2) Obtain a mixture of metal carbonates by $NH_{3(aq.)}$ and $Na_2CO_3$ at pH = 7.5 and 60 °C and holding for 12 h; (3) Calcine at 500 °C for 5 h; (4) Grind with $Li_2CO_3$ and sinter at 500 °C for 5 h and at 900 °C for 12 h; | 163 mAh g$^{-1}$ at 0.1 C; 94% capacity retention after 50 cycles at 1 C; | [148] |
| $LiNi_xMn_yCo_zO_2$ | (1) Adjust metal ratios by acetate precursors; (2) Add $NH_{3(aq.)}$ at pH = 7 and obtain gel precursor at 80 °C; (3) Calcine at 450 °C for 5 h and sinter at 900 °C for 12 h; | 151 mAh g$^{-1}$ at 0.2 C; 84% capacity retention after 150 cycles at 0.2 C; | [145] |
| $Li_{1.2}Ni_{0.13}Mn_{0.54}Co_{0.13}O_2$ | (1) Adjust metal ratios by acetate precursors; (2) Precipitate metals by oxalic acid; (3) Heat hydrothermally at 200 °C for 8 h; (4) Calcine at 450 °C for 5 h and sinter 900 °C for 12 h; | 237 mAh g$^{-1}$ at 0.5 C; 77% capacity retention after 50 cycles at 0.5 C; | [252] |
| **Regeneration** | | | |
| $LiCoO_2$ | Hydrothermally treated cathode in 5.0 M LiOH at 200 °C for 20 h; | 144 mAh g$^{-1}$ at 0.2 C; 92% capacity retention after 40 cycles at 0.2 C; | [246] |
| $LiNi_{0.5}Co_{0.2}Mn_{0.3}O_2$ | (1) Cathode scraps heated at 400 °C for 6 h to burn out acetylene black; (2) Adjust metal ratio (5% Li excess) with lithium acetate; (3) Calcine at 500 °C for 5 h and at 900 °C for 12 h; | 164 mAh g$^{-1}$ at 0.1 C; 89% capacity retention after 100 cycles at 1 C; | [243] |
| $LiNi_{0.5}Co_{0.2}Mn_{0.3}O_2$ | (1) Calcination at 550 °C for 4 h to remove binder and carbon black; (2) Regeneration by LiOH (14% excess); | 161 mAh g$^{-1}$ at 0.1 C; 95% capacity retention after 50 cycles at 0.5 C; | [245] |
| $LiNi_xCo_yMn_zO_2$ | (1) Separation of cathode active material from Al substrate, binder, and carbon by means of dimethyl carbonate and NMP; (2) Cathode is added to 4 M LiOH and autoclaved at 220 °C for 4 h; (3) Sintering with 5% $Li_2CO_3$ excess in $O_2$ atmosphere at 850 °C for 4 h; | 157 mAh g$^{-1}$ at 1 C; 79% capacity retention after 100 cycles at 1 C; | [244] |
| $LiNi_{1/3}Co_{1/3}Mn_{1/3}O_2$ | (1) Cathode material is separated from Al substrate, binder, and carbon by calcination; (2) Ball milling with $Li_2CO_3$ at Li/Me ratio of 1.2 and 500 rpm for 4 h; (3) Sintering at 800 °C for 10 h; | 165 mAh g$^{-1}$ at 0.2 C; 80% capacity retention after 100 cycles at 0.2 C; | [247] |
| $LiCoO_2$ | (1) Cathode material is immersed in 4 M LiOH+KOH solution; (2) Electrochemical dissolution and subsequent precipitation of $LiCoO_2$ at 1 mA cm$^{-2}$ and 40–100 °C; | 135 mAh g$^{-1}$ at 0.2 C; 97% capacity retention after 50 cycles at 0.2 C; | [248] |
| $LiCoO_2$ | (1) Spent LCO powder is placed with 2 M LiOH at S/L = 14; (2) Mixture is heated at 120 °C and sonicated at 1 kW for 10 h; | 131 mAh g$^{-1}$ at 0.2 C; 98% capacity retention after 20 cycles at 0.2 C; | [252] |

## 3. Discussion and Perspective

Above, we have considered the existing LIBs recycling routines—from the cell discharging to the separation and precipitation of the battery-grade compounds. While the methods of a direct cathodes restoration are briefly discussed, a large attention is devoted to the principles of pyro- and hydrometallurgy. Yet, these are more universal and effective approaches.

To sum up, there is several factors to consider when selecting a suitable LIB recycling method. For a concluding remark, we list the main factors in Table 4, where one can distinguish the qualities inherent to pyro- and hydrometallurgy. Specifically, we compare the recycling efficiency, energy consumption, duration, environmental impact, reagents reuse capability, and cost of the process for the pyro- and hydrometallurgy (in-/organic acid, alkaline, and bioleaching). The common tendency observed in Table 4 is an evident trade-off between methods' efficiency (yield, process duration) and sustainability (environmental impact, energy consumption, reagents' reuse). Indeed, pyrometallurgy and cathodes leaching with inorganic acids imply high metal recovery efficiency, but they are not favorable due to either high gaseous emission, or unsustainable reagents. On the contrary, organic acid, alkaline, and bioleaching demonstrate lower efficiency but a benign ecological footprint. Particularly interesting specifics are described for the bioleaching: the time of leaching driven by bacteria can take up to weeks; the spent bacteria-contained solution loses its leaching properties after the first use.

**Table 4.** Qualitative performance comparison of the common LIBs recycling approaches.

| Method | Efficiency | Energy Consumption | Duration | Environmental Impact | Reuse of Reagents | Cost |
|---|---|---|---|---|---|---|
| Pyrometallurgy | ++++ | + | ++ | + | − | + |
| Inorganic acid leaching | ++++ | +++ | ++++ | ++ | ++ | ++ |
| Organic acid leaching | +++ | ++ | +++ | +++ | ++++ | +++ |
| Alkaline leaching | ++ | ++ | ++ | +++ | ++++ | +++ |
| Bioleaching | +++ | ++ | + | ++++ | + | ++ |

+: worst performance; ++++: best performance; −: not applicable.

With the rapid development of the EV and stationary energy storage markets, the demand for high-energy and high-power LIBs will continue growing in the next decades. Therefore, exploring efficient cathode recycling processes is essential for the further sustainable development of the LIBs industry. The global authorities forecast both extensive and intensive growth of the recycling infrastructure. By 2022, 32 kT year$^{-1}$ recycling capacity has been already installed, whereas an additional 70 kT are planned for this decade to be set worldwide [29]. Besides new recycling gigafactories, novel technologies should be developed and introduced to boost the metals recovery efficiency and minimize the cost of the process.

Emerging LIB cell pretreatment technologies are expected to crucially influence future recycling. For instance, in the case of hydrometallurgy, automatic cell dismantling would improve the safety and efficiency of the whole recycling process by minimizing active material losses and enhancing selectivity.

Selective metal leaching is another promising method to improve overall efficiency. It allows to extract specific metals, e.g., Co, from spent mixed cathodes or multi-component scrap leaving less valuable or toxic components in the residue. The economical and time issues of the NMC cathodes recycling would be overtaken significantly with this approach. Leaching organic acids, in addition to their eco-friendliness, also bear several other benefits. They can be used as sol-gel precursors in the cathode restoration processes and serve as precipitation and reducing agents.

The alternative concepts of cathode material resynthesis and its direct regeneration might be simpler and more cost-efficient for the LIBs production industry compared to the hydrometallurgical methods. Both approaches would deprive the recycling process with the complex steps of metal ions separation, extraction, and salt precipitation. The

main issue of the direct regeneration to overtake is a unification of the methodology to be applicable for a large variety of rapidly exploring types of cathode materials.

In the end, we suppose it is necessary to summarize the main obstacles, promising solutions, and future targets for LIBs recycling. The current challenges can be conditionally separated into three groups: technological, geographical, and economic (Figure 7). While the technology-related issues (e.g., sustainable aspects, unification recycling as well as ineffective metal separation) were thoroughly discussed in the current review, the geography and economy problems were just superficially addressed. Overall, attentive control over the amount of accumulating EoL LIBs, lobbying the recycling regulations, tuning raw/recycled materials supply chain for LIBs production, and promoting other circular economy principles [26] would complete transforming LIB recycling into an environmentally benign and cost-efficient procedure. Finally, a high volume (>80%) of LIB recycling would indemnify cells manufacturing the effects of sudden raw material supply shortages and reduce the total cost of LIBs production.

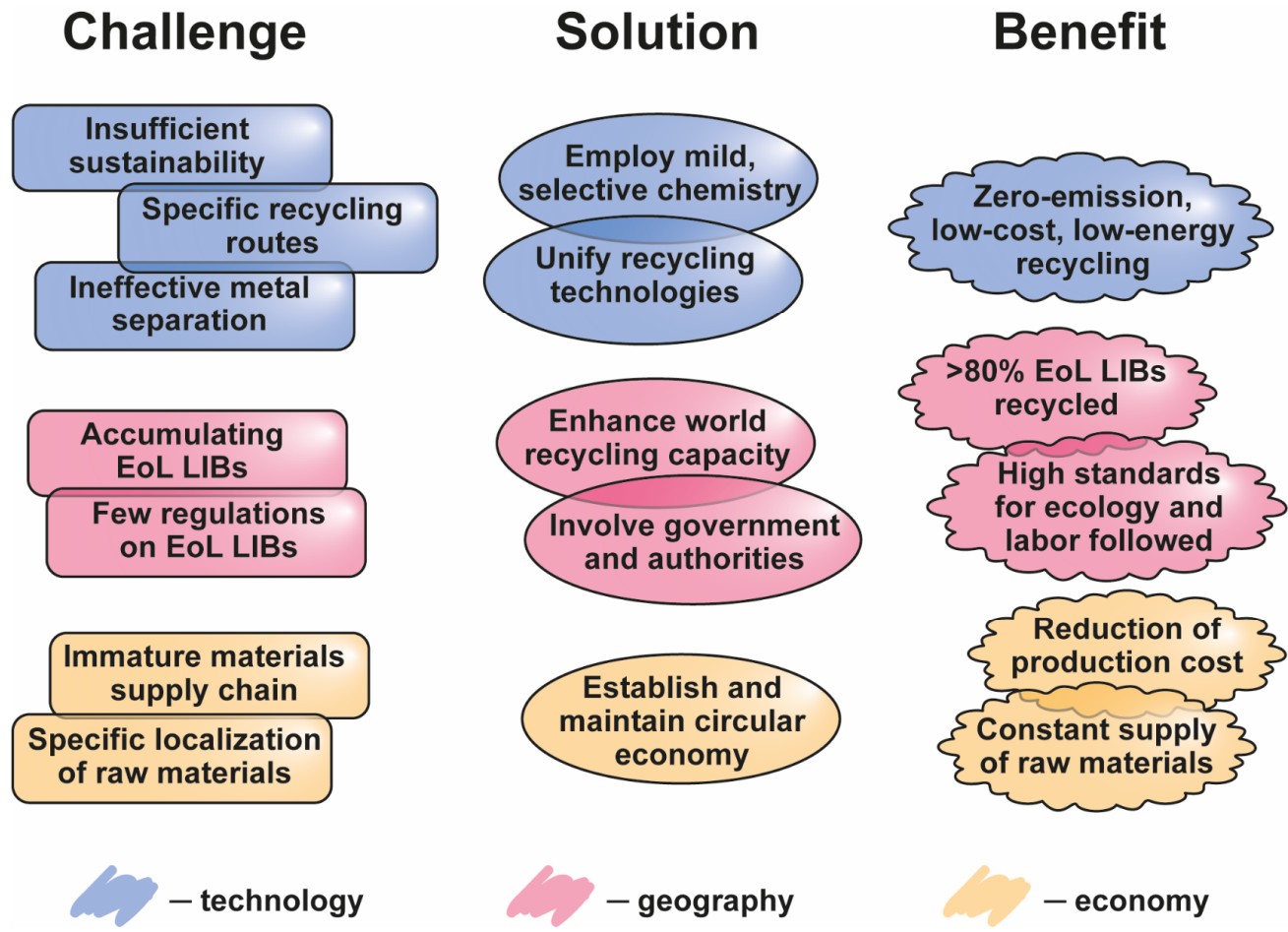

**Figure 7.** Perspective roadmap of EoL LIB recycling development illustrating the existing challenges, proposing solutions, and benefits that can be gained as a result of solutions implementation.

## 4. Conclusions

Indeed, the LIB market has been expanding at an exponential rate, owing to the proliferation of portable devices and, more recently, the development of electric vehicles. Taking into account such a rapid rise, a large mass of EoL LIBs is accumulating and needs to be ecologically utilized (recycled) to minimize environmental damage. Despite some already existing recycling facilities, many of them still use high-temperature processes (pyrometallurgy) that emit a bunch of hazardous gases and produce low-purity products (slag, alloys). In addition, the current production rate of LIBs significantly exceeds their recycling capacity. All of that stimulates fundamental and applied investigations for efficient LIB utilization as well as the extensive growth of recycling facilities (plants). Nowadays, the most prospective recycling technologies are likely associated with hydrometallurgy. Despite the fact that hydrometallurgical recycling is only entering the industry at a lab scale, it has great potential due to its large energy savings, lower $CO_2$ footprint, and much higher purity of the recycled material compared to traditional pyrometallurgy. Relying on optimizing the LIBs pretreatment (discharging, dismantling, and sorting), the novel recycling approaches would drastically diminish the pollution and dependence on raw material extraction and mining for LIBs production. When the transition to mixed cathode materials (rather than a single, co-contained) is completed, hydrometallurgy will most likely be a dominant recycling pathway in the near future.

In the current review, various LIBs' pretreatment techniques (Section 2.1: manual dismantling; crushing and sieving, etc.) were considered, as were different cathode leaching methods (Section 2.2.2: inorganic, organic acids; ammonia; bioleaching); and separation technologies (Section 2.2.2: extraction; precipitation). In fact, at each recycling step, a certain procedure has its pros and cons. For example, inorganic acids are more efficient in leaching than their organic counterparts—the former require lower acid concentrations and cathode-liquid ratios to achieve complete cathode material dissolution. At the same time, some organic acids are more eco-friendly, more affordable, and safer. Alkaline leaching and bioleaching are some of the currently emerging recycling technologies, although they still suffer from metal separation issues and too-long procedure durations, respectively. It is curious to observe, how much research the non-traditional approaches have accumulated (Section 2.2.3: electrochemical treatment; utilization of ionic liquids, deep eutectic solvents, etc.), which makes them potentially competitive with the conventional hydrometallurgical recycling routes. The direct cathode restoration technologies (Section 2.3) might also be applicable, as they allow a drastic decrease in the number of recycling stages—they resynthesize the cathode right from the leaching solution or directly lithiate the spent active material. However, the restoration efficiency is highly sensitive to the temperature regime, media, and phase purity of the final product, which impedes scaling the direct restoration of cathodes to an industrial level.

Here, we have considered plenty of currently available industrial and emerging lab-scale technologies for recycling the cathodes of LIBs. There are indeed several sustainable recycling processes in development. Therefore, much research now should be devoted to accelerating their introduction to the next, industrial stage. We hope that the fundamental and applied knowledge described in this report will be useful in the future development of an environmentally friendly circular economy and, in particular, the LIBs recycling industry.

**Author Contributions:** Conceptualization, N.A. and A.M.; methodology, N.A.; formal analysis, N.A., A.S.A.-Q. and A.M.; investigation, N.A.; data curation, A.M. and A.S.A.-Q. writing—original draft preparation, N.A.; writing—review and editing, A.M. and A.S.A.-Q.; visualization, N.A.; supervision, A.M. All authors have read and agreed to the published version of the manuscript.

**Funding:** This research received no external funding.

**Data Availability Statement:** Data sharing not applicable.

**Acknowledgments:** The authors gratefully acknowledge the support and scientific discussions with Mustafa AlAli.

**Conflicts of Interest:** The authors declare no conflict of interest.

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
