# Peer review of "Li-Ion Battery Cathode Recycling: An Emerging Response to Growing Metal Demand and Accumulating Battery Waste"

_electronics, doi:10.3390/electronics12051152_

Round 1
Reviewer 1 Report
Dear Editor, Electronics
Thank you for sending me the manuscript for review. The manuscript seems a good review on Li-ion battery recycling. My recommend is to include and discuss the recycling achieving for low-temperature Li-ion battery in a subheading.
I think the manuscript discusses advances in Li-ion Batteries. The overall presentation, figures, conclusions and reference sections are fine. However, I suggested the author include a topic on low temperature Li-ion batteries and discuss their recycling process.
The manuscript may be considered for publication after revision.
Md Abdus Subhan
SUST
Sylhet
Bangladesh
Author Response
In this review, we from the chemical point of view analyze the LIBs utilization technologies with a particular emphasis on cathode recycling. The current paper mainly reviews the most recent recycling trends (2020-2022). Throughout the review, we follow the logic to pay attention to every recycling stage in a universal manner and consider every technological option for each stage: weather it seems outdated (discuss briefly) or emerging (provide more details and analysis). Such a broad and topical coverage distinguishes this review from the other ones, which focus separately on either the recycling of EV cathodes [24], LiCoO2 cathodes [25], and all LIBs’ components [26,27], pretreatment stage of the recycling [28], or the commercialized technologies [29]. In other words, we suggest this review as a self-sustain, independent study that is enough for a primary and sufficient immersion into the LIB cathode recycling field. Indeed, this review covers the whole spectrum of the cathode recycling processes starting from cell discharging and finishing battery-grade salt extraction. After an introduction (environmental impact, existing recycling processes, section 1), we focus strictly on the LIB’s processing methodology and the chemistry used within — discuss the recycling structure, provide specific conditions and reagents, and analyze the advantages and limitations of each method (section 2). Reasonable attention is devoted to the hydrometallurgical branch of recycling as one of the most promising routes for the end-of-life (EoL) LIBs treatment. Further, we proceed through the latest cathode restoration techniques and, finally, finish with a summary and perspective for the LIB cathode recycling. With the current review article, we would like to provide interested scientists, especially, novices to the LIB recycling culture, with valuable and hands-on both theoretical and practical knowledge. We hope that the LIBs recycling technological aspects provided and discussed in this paper will enhance the search and then implementation of novel sustainable recycling routes.

Reviewer 2 Report
This paper introduced analysis and discussion of different LIB recycling approaches, emphasizing cathode processing. The technological aspects of LIB's pretreatment, sorting and dissolving the cathode, separation of leached elements, and obtaining high-purity materials have been discussed. The proven and emerging approaches and compare pyrometallurgy, hydrometallurgy, and cathode’s direct restoration methods have been analyzed.
This paper is well-written and it has valuable information, it can be accepted for publication after considering the following points:
1- The contents, logic, and motivation should be compared with other review papers introduced in the literature. Out of this comparison, the authors can clearly present the main contribution and innovation involved in this paper which is not shown clearly in this paper.
2- The author should provide a table that can compare the pyrometallurgy, hydrometallurgy, and hybrid material extraction procedures from different aspects such as cost, environmental impact, separation time, energy consumption, etc., and provide a general conclusion about their feasibility.
3- The authors should provide the expected future trends in this important issue.
4- The authors should include a section that can help researchers, designers, investors, and decision-makers that intend to work in their area about the challenges, obstacles, and benefits of using the EOL batteries separation projects worldwide.

Author Response
Authors’ responses to the Reviewers’ comments (also in attached PDF, for convenience)
SUMMARY
The authors would like to express their gratitude to all the reviewers involved in the process of improving our manuscript. We appreciate all the remarks regarding the text clarification, stylistic and logic issues, and the article’s proper assembling. We hope all the concerns are now fully addressed and thoroughly elucidated.
Here, the reviewers’ commentaries are numbered and enclosed after a vertical bold line. The authors’ responses are located right after the reviewers’ comments. Portions of the response text are bolded for emphasis. Excerpts from the first manuscript submission are presented in blue font. Changes made to the revised manuscript text in response to reviewer’s comments have been highlighted here and in the revised manuscript.
|
REVIEWER #2 |
|
This paper introduced analysis and discussion of different LIB recycling approaches, emphasizing cathode processing. The technological aspects of LIB's pretreatment, sorting and dissolving the cathode, separation of leached elements, and obtaining high-purity materials have been discussed. The proven and emerging approaches and compare pyrometallurgy, hydrometallurgy, and cathode’s direct restoration methods have been analyzed. This paper is well-written and it has valuable information, it can be accepted for publication after considering the following points: |
We are thankful to the Reviewer for his/her high appreciation of our work. In the following, we have addressed the points and implemented the necessary changes in the manuscript.
|
COMMENTARY 1 |
|
The contents, logic, and motivation should be compared with other review papers introduced in the literature. Out of this comparison, the authors can clearly present the main contribution and innovation involved in this paper which is not shown clearly in this paper. |
The authors thank the Reviewer for such a valuable request to emphasize the practical difference between our review from others devoted to Li-ion batteries (LIBs) recycling. Accordingly, we rephrased the main text (Pages 2-3, Lines 79-104). All our modifications were tracked using track&changes function.
In this review, we from the chemical point of view analyze the LIBs utilization technologies with a particular emphasis on cathode recycling. The current paper mainly reviews the most recent recycling trends (2020-2022). Throughout the review, we follow the logic to pay attention to every recycling stage in a universal manner and consider every technological option for each stage: weather it seems outdated (discuss briefly) or emerging (provide more details and analysis). Such a broad and topical coverage distinguishes this review from the other ones, which focus separately on either the recycling of EV cathodes [24], LiCoO2 cathodes [25], and all LIBs’ components [26,27], pretreatment stage of the recycling [28], or the commercialized technologies [29]. In other words, we suggest this review as a self-sustain, independent study that is enough for a primary and sufficient immersion into the LIB cathode recycling field. Indeed, this review covers the whole spectrum of the cathode recycling processes starting from cell discharging and finishing battery-grade salt extraction. After an introduction (environmental impact, existing recycling processes, section 1), we focus strictly on the LIB’s processing methodology and the chemistry used within — discuss the recycling structure, provide specific conditions and reagents, and analyze the advantages and limitations of each method (section 2). Reasonable attention is devoted to the hydrometallurgical branch of recycling as one of the most promising routes for end-of-life (EoL) LIBs treatment. Further, we proceed through the latest cathode restoration techniques and, finally, finish with a summary and perspective for the LIB cathode recycling. With the current review article, we would like to provide interested scientists, especially, novices to the LIB recycling culture, with valuable and hands-on both theoretical and practical knowledge. We hope that the LIBs recycling technological aspects provided and discussed in this paper will enhance the search and then implementation of novel sustainable recycling routes.
|
COMMENTARY 2 |
|
The author should provide a table that can compare the pyrometallurgy, hydrometallurgy, and hybrid material extraction procedures from different aspects such as cost, environmental impact, separation time, energy consumption, etc., and provide a general conclusion about their feasibility. |
Inspired by the Commentaries 2-4 of the Reviewer, we decided to introduce a separate part to our manuscript (section 3 — Discussion and Perspective; pages 31-33, lines 1098-1161), where we compare the specified recycling approaches, discuss perspective, and provide readers with exact advice in the recycling field.
Particularly, the authors agree with the Reviewer’s suggestion to compare the best recycling technologies among discussed (pyrometallurgy and hydrometallurgy family) from different aspects (pages 31-32, lines 1099-1119). All our changes were tracked.
Above, we have considered the existing LIBs recycling routines — from the cell discharging to the separation and precipitation of the battery-grade compounds. While the methods of direct cathode restoration are briefly discussed, a large attention is devoted to the principles of pyro- and hydrometallurgy. Yet, these are more universal and effective approaches.
To sum up, there are several factors to consider when selecting a suitable LIB recycling method. For a concluding remark, we list the main factors in Table 4, where one can distinguish the qualities inherent to pyro- and hydrometallurgy. Specifically, we compare the recycling efficiency, energy consumption, duration, environmental impact, reagents reuse capability, and cost of the process for the pyro- and hydrometallurgy (in-/organic acid, alkaline, and bioleaching). The common tendency observed in Table 4 is an evident trade-off between methods’ efficiency (yield, process duration) and sustainability (environmental impact, energy consumption, reagents’ reuse). Indeed, pyrometallurgy and cathodes leaching with inorganic acids imply high metal recovery efficiency, but they are not favorable due to either high gaseous emission, or unsustainable reagents. On the contrary, organic acid, alkaline, and bioleaching demonstrate lower efficiency but a benign ecological footprint. Particularly interesting specifics are described for the bioleaching: the time of leaching driven by bacteria can take up to weeks; the spent bacteria-contained solution loses its leaching properties after the first use.
Table 4. Qualitative performance comparison of the common LIBs recycling approaches.
|
Method |
Efficiency |
Energy consumption |
Duration |
Environmental impact |
Reuse of reagents |
Cost |
|
Pyrometallurgy |
++++ |
+ |
++ |
+ |
− |
+ |
|
Inorganic acid leaching |
++++ |
+++ |
++++ |
++ |
++ |
++ |
|
Organic acid leaching |
+++ |
++ |
+++ |
+++ |
++++ |
+++ |
|
Alkaline leaching |
++ |
++ |
++ |
+++ |
++++ |
+++ |
|
Bioleaching |
+++ |
++ |
+ |
++++ |
+ |
++ |
+: worst performance; ++++: best performance; −: not applicable
|
COMMENTARY 3 |
|
The authors should provide the expected future trends in this important issue. |
We appreciate this valuable Reviewer’s comment requesting to discuss the future trends and perspective for LIB recycling. Below, we introduce the corresponding discussion to the main text (page 32, lines 1121-1146).
With the rapid development of the EV and stationary energy storage markets, the demand for high-energy and high-power LIBs will continue growing in the next decades. Therefore, exploring efficient cathode recycling processes is essential for the further sustainable development of the LIBs industry. The global authorities forecast both extensive and intensive growth of the recycling infrastructure. By 2022, 32 ktons year-1 recycling capacity has been already installed, whereas additional 70 ktons are planned for this decade to be set worldwide [1]. Besides new recycling gigafactories, novel technologies should be developed and introduced to boost the metals recovery efficiency and minimize the cost of the process.
Emerging LIB cell pretreatment technologies are expected to crucially influence future recycling. For instance, in the case of hydrometallurgy automatic cell dismantling would improve the safety and efficiency of the whole recycling process by minimizing active material losses and enhancing selectivity.
Selective metal leaching is another promising method to improve overall efficiency. It allows to extract specific metals, e.g., Co, from spent mixed cathodes or multi-component scrap leaving less valuable or toxic components in the residue. The economical and time issues of the NMC cathodes recycling would be overtaken significantly with this approach. Leaching organic acids, in addition to their eco-friendliness, also bear several other benefits. They can be used as sol-gel precursors in the cathode restoration processes and serve as precipitation and reducing agents.
The alternative concepts of cathode material resynthesis and its direct regeneration might be simpler and more cost-efficient for the LIBs production industry compared to the hydrometallurgical methods. Both approaches would deprive the recycling process with the complex steps of metal ions separation, extraction, and salt precipitation. The main issue of direct regeneration to overtake is the unification of the methodology to be applicable for a large variety of rapidly exploring types of cathode materials.
[1] Baum, Z.J., Bird, R.E., Yu, X., & Ma, J. (2022). Lithium-Ion Battery Recycling ─ Overview of Techniques and Trends. ACS Energy Lett. 7(2), 712–719.
|
COMMENTARY 4 |
|
The authors should include a section that can help researchers, designers, investors, and decision-makers that intend to work in their area about the challenges, obstacles, and benefits of using the EOL batteries separation projects worldwide. |
The authors would like to thank the Reviewer for the great idea to include discussing the challenges, solutions, and benefits of LIBs recycling. Below, we carefully address this request and introduce Figure 7 and the corresponding discussion in the manuscript (pages 32-33, lines 1147-1161).
In the end, we suppose it is necessary to summarize the main obstacles, promising solutions, and future targets for LIBs recycling. The current Challenges can be conditionally separated into three groups: technological, geographical, and economic (Figure 7). While the technology-related issues (e.g., sustainable aspects, unification recycling as well as ineffective metal separation) were thoroughly discussed in the current review, the geography and economy problems were just superficially addressed. Overall, attentive control over the amount of accumulating EoL LIBs, lobbying the recycling regulations, tuning raw/recycled materials supply chain for LIBs production, and promoting other circular economy principles [1] would complete transforming LIB recycling into an environmentally benign and cost-efficient procedure. Finally, a high volume (>80%) of LIB recycling would indemnify cells manufacturing the effects of sudden raw material supply shortages and reduce the total cost of LIBs production.
Figure 7. Perspective roadmap of EoL LIB recycling development illustrating the existing Challenges, proposing Solutions, and Benefits that can be gained as a result of solutions implementing.
[1] Piątek, J., Afyon, S., Budnyak, T. M., Budnyk, S., Sipponen, M. H., & Slabon, A. (2021). Sustainable Li‐ion batteries: chemistry and recycling. Advanced Energy Materials, 11(43), 2003456.

Reviewer 3 Report
Li-ion battery cathode recycling: an emerging response to growing metal demand and accumulating battery waste is an interesting review by the authors. I want to recommend for its publication after a few minor issues are fixed:
1. The authors should write clearly how this review work is different in terms of novelty and why this review is needed more deeply in the introduction.
2. I cannot find what the authors think about a framework for developing future recycling infrastructures.
3. The authors should write and discuss the topic of LIB recycling differentiation between academic approaches and industrial reality.
4. Some of the important references must be discussed: Sustainable Materials and Technologies 32, e00396, 2022 and Journal of Cleaner Production 384, 135520, 2023
5. The authors should make a picture from one idea to make the readers understand LB recycling, its challenges, and what are possible ways to overcome them.
6. English grammatical errors need to be corrected throughout.
Author Response
Authors’ responses to the Reviewers’ comments (also in PDF attached)
SUMMARY
The authors would like to express their gratitude to all the reviewers involved in the process of improving our manuscript. We appreciate all the remarks regarding the text clarification, stylistic and logical issues, and the article’s proper assembling. We hope all the concerns are now fully addressed and thoroughly elucidated.
Here, the reviewers’ commentaries are numbered and enclosed after a vertical bold line. The authors’ responses are located right after the reviewers’ comments. Portions of the response text are bolded for emphasis. Excerpts from the first manuscript submission are presented in a blue font. Changes made to the revised manuscript text in response to the reviewer’s comments have been highlighted here and in the revised manuscript.
|
REVIEWER #3 |
|
Li-ion battery cathode recycling: an emerging response to growing metal demand and accumulating battery waste is an interesting review by the authors. I want to recommend for its publication after a few minor issues are fixed: |
We appreciate the Reviewer for the careful evaluation of our work in the field of Li-ion batteries recycling processes. Below, we tried to address all the mentioned commentaries and support them with the changes in the text.
|
COMMENTARY 1 |
|
The authors should write clearly how this review work is different in terms of novelty and why this review is needed more deeply in the introduction. |
The authors thank the Reviewer for such a valuable request to emphasize the practical difference of our review from the others devoted to Li-ion batteries (LIBs) recycling. Accordingly, we rephrased the main text (Pages 2-3, Lines 79-104):
In this review, from the chemical point of view, we analyze the LIBs utilization technologies with a particular emphasis on the cathode recycling. The current paper mainly reviews the most recent recycling trends (2020-2022). Throughout the review, we follow the logic to pay attention to every recycling stage in a universal manner and consider every technological option for each stage: weather it seems outdated (discuss briefly) or emerging (provide more details and analysis). Such a broad and topical coverage distinguishes this review from the other ones, which focus separately on either the recycling of EV cathodes [1], LiCoO2 cathodes [2], and all LIBs’ components [3-4], pretreatment stage of the recycling [5], or the commercialized technologies [6]. In other words, we suggest this review as a self-sustain, independent study that is enough for a primary and sufficient immersion into the LIB cathode recycling field. Indeed, this review covers the whole spectrum of the cathode recycling processes starting from cell discharging and finishing with battery-grade salts extracting. After an introduction (environmental impact, existing recycling processes, section 1), we focus strictly on the LIB’s processing methodology and the chemistry used within — discuss the recycling structure, provide specific conditions and reagents, and analyze advantages and limitations of each method (section 2). Reasonable attention is devoted to the hydrometallurgical branch of recycling as one of the most promising routes for the end-of-life (EoL) LIBs treatment. Further, we proceed through the latest cathode restoration techniques and, finally, finish with a summary and perspective for the LIB cathode recycling. With the current review article, we would like to provide interested scientists, especially, novices to the LIB recycling culture, with valuable and hands-on both theoretical and practical knowledge. We hope that the LIBs recycling technological aspects discussed in this paper will enhance the search and then implementation of novel sustainable recycling routes.
[1] Or, T., Gourley, S. W., Kaliyappan, K., Yu, A., & Chen, Z. (2020). Recycling of mixed cathode lithium‐ion batteries for electric vehicles: Current status and future outlook. Carbon Energy, 2(1), 6-43.
[2] Botelho Junior, A. B., Stopic, S., Friedrich, B., Tenório, J. A. S., & Espinosa, D. C. R. (2021). Cobalt recovery from Li-ion battery recycling: a critical review. Metals, 11(12), 1999.
[3] Piątek, J., Afyon, S., Budnyak, T. M., Budnyk, S., Sipponen, M. H., & Slabon, A. (2021). Sustainable Li‐ion batteries: chemistry and recycling. Advanced Energy Materials, 11(43), 2003456.
[4] Wang, Y., An, N., Wen, L., Wang, L., Jiang, X., Hou, F., ... & Liang, J. (2021). Recent progress on the recycling technology of Li-ion batteries. Journal of Energy Chemistry, 55, 391-419.
[5] Kim, S., Bang, J., Yoo, J., Shin, Y., Bae, J., Jeong, J., ... & Kwon, K. (2021). A comprehensive review on the pretreatment process in lithium-ion battery recycling. Journal of Cleaner Production, 294, 126329.
[6] Baum, Z. J., Bird, R. E., Yu, X., & Ma, J. (2022). Lithium-ion battery recycling ─ overview of techniques and trends. ACS Energy Lett. 2022, 7, 2, 712–719.
|
COMMENTARY 2 |
|
I cannot find what the authors think about a framework for developing future recycling infrastructures. |
We appreciate this valuable Reviewer’s comment requesting to discuss the future trends and perspective for LIB recycling. Below, we introduce the corresponding discussion to the main text (page 32, lines 1121-1146).
With the rapid development of the EV and stationary energy storage markets, the demand for high-energy and high-power LIBs will continue growing in the next decades. Therefore, exploring efficient cathode recycling processes is essential for the further sustainable development of the LIBs industry. The global authorities forecast both extensive and intensive growth of the recycling infrastructure. By 2022, 32 ktons year-1 recycling capacity has been already installed, whereas additional 70 ktons are planned for this decade to be set worldwide [1]. Besides new recycling gigafactories, novel technologies should be developed and introduced to boost the metals recovery efficiency and minimize the cost of the process.
Emerging LIB cell pretreatment technologies are expected to crucially influence future recycling. For instance, in the case of hydrometallurgy automatic cell dismantling would improve the safety and efficiency of the whole recycling process by minimizing active material losses and enhancing selectivity.
Selective metal leaching is another promising method to improve overall efficiency. It allows to extract specific metals, e.g., Co, from spent mixed cathodes or multi-component scrap leaving less valuable or toxic components in the residue. The economical and time issues of the NMC cathodes recycling would be overtaken significantly with this approach. Leaching organic acids, in addition to their eco-friendliness, also bear several other benefits. They can be used as sol-gel precursors in the cathode restoration processes and serve as precipitation and reducing agents.
The alternative concepts of cathode material resynthesis and its direct regeneration might be simpler and more cost-efficient for the LIBs production industry compared to the hydrometallurgical methods. Both approaches would deprive the recycling process with the complex steps of metal ions separation, extraction, and salt precipitation. The main issue of direct regeneration to overtake is the unification of the methodology to be applicable for a large variety of rapidly exploring types of cathode materials.
[1] Baum, Z.J., Bird, R.E., Yu, X., & Ma, J. (2022). Lithium-Ion Battery Recycling ─ Overview of Techniques and Trends. ACS Energy Lett. 7(2), 712–719.
|
COMMENTARY 3 |
|
The authors should write and discuss the topic of LIB recycling differentiation between academic approaches and industrial reality. |
The authors express their gratitude to the Reviewer for the idea to emphasize the differences between lab-scale and industrial features of LIB recycling. Below, we have addressed this issue in more detail (page 9, lines 333-341).
Much attention will be devoted to the designed routines, process conditions, and chemicals used by leading research groups specialized in LIBs recycling over the last decade. Looking ahead, it is important to note there are recycling steps already suitable for use in industry (e.g., pyrometallurgy; cell crushing; sieving, and separation of crushed fractions), other methods are currently adopting to a large scale (inorganic acid leaching; metal ions separation; salt precipitation), whereas the rest part is emerging and capable only with the laboratory conditions (organic, alkaline, bioleaching; cell dismantling; cathodes direct regeneration; etc.). Throughout this section, we emphasize a correlation between an exact technology of a certain recycling stage and the acceptable application level, as well as discuss requirements and perspectives for the technology to be introduced into the industry.
|
COMMENTARY 4 |
|
Some of the important references must be discussed: Sustainable Materials and Technologies 32, e00396, 2022 and Journal of Cleaner Production 384, 135520, 2023. |
The authors appreciate the Reviewer’s suggestion and are thankful for providing us with highly relevant publications in the area of LIB recycling applications and general energy storage. We have introduced the proposed citations into the manuscript (page 2, lines 58).
It was not a surprise that over the last ten years, the demand for LIBs has drastically increased [1-2].
[1] Bejigo, K.S., Natarajan, S., Bhunia, K., Elumalai, V., & Kim, S.J. (2023). Recycling of value-added products from spent lithium-ion batteries for oxygen reduction and methanol oxidation reactions. Journal of Cleaner Production, 384, 135520.
[2] Sahu, M., Hajra, S., Jadhav, S., Panigrahi, B. K., Dubal, D., & Kim, H. J. (2022). Bio-waste composites for cost-effective self-powered breathing patterns monitoring: An insight into energy harvesting and storage properties. Sustainable Materials and Technologies, 32, e00396.
|
COMMENTARY 5 |
|
The authors should make a picture from one idea to make the readers understand LB recycling, its challenges, and what are possible ways to overcome them. |
The authors would like to thank the Reviewer for the great idea to include discussing the challenges, possible ways for overcoming them, as well as their practical impact on the LIBs recycling outcomes. Below, we carefully address this request and introduce Figure 7 and the corresponding discussion in the manuscript (pages 32-33, lines 1147-1161.
In the end, we suppose it is necessary to summarize the main obstacles, promising solutions, and future targets for LIBs recycling. The current Challenges can be conditionally separated into three groups: technological, geographical, and economic (Figure 7). While the technology-related issues (e.g., sustainable aspects, unification recycling as well as ineffective metal separation) were thoroughly discussed in the current review, the geography and economy problems were just superficially addressed. Overall, attentive control over the amount of accumulating EoL LIBs, lobbying the recycling regulations, tuning raw/recycled materials supply chain for LIBs production, and promoting other circular economy principles [1] would complete transforming LIB recycling into an environmentally benign and cost-efficient procedure. Finally, a high volume (>80%) of LIB recycling would indemnify cells manufacturing the effects of sudden raw material supply shortages and reduce the total cost of the LIBs production.
Figure 7. Perspective roadmap of EoL LIB recycling development illustrating the existing Challenges, proposing Solutions, and Benefits that can be gained as a result of solutions implementation.
[1] Piątek, J., Afyon, S., Budnyak, T. M., Budnyk, S., Sipponen, M. H., & Slabon, A. (2021). Sustainable Li‐ion batteries: chemistry and recycling. Advanced Energy Materials, 11(43), 2003456.
|
COMMENTARY 6 |
|
English grammatical errors need to be corrected throughout. |
We thank the Reviewer for the kind request to check the English language throughout the review. Accordingly, we introduced the grammatical improvements as much as we could.n All changes are highlighted and tracked.

Round 2
Reviewer 1 Report
Dear Editor, Electronics
Thank you for sending me the manuscript again for review. Authors have improved some parts in revised version. However, the response regarding my recommendation was not relevant/satisfactory. Recycling of spent LiB battery cathodes for low-temperature use is an important area. I recommend to address this issue in the review.
Sincerely
Reviewer
Author Response
|
Thank you for sending me the manuscript for review. The manuscript seems a good review on Li-ion battery recycling. My recommend is to include and discuss the recycling achieving for low-temperature Li-ion battery in a subheading. I think the manuscript discusses advances in Li-ion Batteries. The overall presentation, figures, conclusions and reference sections are fine. The manuscript may be considered for publication after revision. |
We would like to thank the reviewer for elaboration of the manuscript and its appreciation. Below, we have addressed the reviewer’s feedback to the manuscript.
|
COMMENTARY 1 |
|
I suggested the author include a topic on low temperature Li-ion batteries and discuss their recycling process. |
Following the Reviewer’s strong recommendation, we added the following section into the manuscript regarding low-temperature Li-ion batteries recycling (pages 10-11, lines 353-390):
Special attention should be devoted to the low-temperature (LT) LIBs. Indeed, at temperatures below 0 °C LIBs generally face severe energy and capacity losses, impeded interfacial ionic transfer, and poor ions diffusion in bulk electrodes. In general, the main problems for LIBs at LTs are associated with the graphite anode (slow Li intercalation, high resistance of SEI) and liquid electrolyte (solidification, increased resistance). The anode-related issues are currently trying to be solved by the graphite oxidizing and metal nanoparticles’ introduction (reduction of lithiation overpotential), particle size reduction, and structure and morphology modification (reduce Li plating). The spectrum of electrolyte improvement solutions is large and implies salt and solvent modification [77–79]. An efficient electrolyte for LT LIBs can be obtained by introducing F-contained chemistry (e.g., LiTFSI salt; fluoroethylene carbonate solvent additive, etc.) that would stabilize the SEI and provide the electrolyte with high ionic conductivity and low viscosity.
Improving cathodes is another path toward LT LIBs development. Currently, besides the enhanced impedance, at LT the cathodes suffer from dissolution and re-deposition of transition metals on the surface of the separator and anode [80]. The common solutions for the LT cathodes are surface coating (reduces contact resistance [81]), metal doping (improves the mobility of Li+, stabilizes cathode-electrolyte interface (CEI) [82]), and particle size reduction (facilitates lithium insertion/extraction [83]). Overall, the modification of existing cathodes for LT LIBs requires a careful design of CEI by tuning the transition metal ratios and cathode surface pretreatment (e.g., by LiF-rich layers). In addition to the cathode modification of common materials (LCO, NMC, LFP), their substitution for emerging ones might be also beneficial for LT LIBs, such as Li3V2(PO4)3, Nb2O5, or Ni-based Prussian blue analogs [84].
Of course, both the cathode modification and the use of completely new materials call for severe modification of the recycling process of LIB as a whole and that of cathode materials, in particular. Such, the LiF introduction requires a responsible deactivation method during the recycling to avoid HF emission. The possible options are either selective capturing of the hazardous F-contained gases or LIB’s components treatment in strong bases. e.g., NaOH [85]. The use of V-based cathodes additionally requires specific leaching conditions for the reductive/oxidative dissolution of vanadium. The sulfuric acid leaching solutions enhanced with a reductive agent (H2O2, Na2SO3) are capable to recover stable V4+ in the form of VOSO4 [86]. The use of expensive transition metals (e.g., Nb) as dopants or a main material bears additional economical challenges to the recycling of such cathodes. The losses of such high-cost materials should be minimized, otherwise, the recycling profitability will be questioned. Overall, the LT LIBs development and, particularly, their recycling technology demands a deeper investigation in the nearest future and deserves a separate comprehensive overview. Meanwhile, in the current study, we are mostly focused on the recycling of traditional cathode materials: LCO and NMC.
LIB recycling can be separated into the “reuse” and “recovery” sub-processes. Such materials as Cu and Al (current collectors), Fe (case), and plastics can be directly reused after disassembly, and we will not devote much attention to them. We associate the “recovery” branch of recycling with processing a cathode material specifically, i.e., transforming the complex black mass to the high-purity chemical reagents. For convenience, we will join cell pretreatment techniques, namely, discharging, disassembling, and detaching active material, and will focus on all of them throughout the review.
